

# Snowfall retrieval at X, Ka and W band: consistency of backscattering and microphysical properties using BAECC ground-based measurements

Marta Tecla Falconi[1], Annakaisa von Lerber[2], Davide Ori[3], Frank Silvio Marzano[1], and Dmitri Moisseev[2,4]

[1]Department of Information Engineering, Sapienza University of Rome, Italy and CETEMPS, L'Aquila, Italy
[2]Finnish Meteorological Institute, Finland
[3]Institute for Geophysics and Meteorology, University of Cologne, Cologne, Germany
[4]Department of Physics, University of Helsinki, Finland

*Correspondence to:* Marta Tecla Falconi (martatecla.falconi@uniroma1.it)

**Abstract.** Radar-based snowfall intensity retrieval is investigated at centimeter and millimeter wavelengths using high-quality collocated ground-based multi-frequency radar and video-disdrometer observations. Using data from four snowfall events, recorded during the Biogenic Aerosols Effects on Clouds and Climate (BAECC) campaign in Finland, measurements of liquid-water-equivalent snowfall rate $S$ are correlated to radar equivalent reflectivity factors $Z_e$, measured by the Atmospheric Radi-

5 ation Measurement (ARM) cloud radars operating at X, Ka and W frequency bands. From these coupled observations power-law $Z_e$-$S$ relationships are derived for all considered frequencies and distinguishing fluffy from rimed snowfall. Interestingly fluffy-snow events show a spectrally distinct signature of $Z_e$-$S$ with respect to rimed-snow cases. In order to understand the connection between snowflake microphysical and multi-frequency backscattering properties, numerical simulations are also performed by using the particle size distribution provided by the in-situ video-disdrometer. The latter are carried out by using

both the T-matrix method (TMM) for soft-spheroids with different aspect ratios and exploiting a pre-computed discrete dipole approximation (DDA) database for complex-shape snowflakes. Based on the presented results, it is concluded that the soft-spheroid approximation can be adopted to explain the observed multi-frequency $Z_e$-$S$ relations if a proper spheroid aspect ratio is selected. The latter may depend on the snowfall type. A further analysis of the backscattering simulations reveals that TMM cross-sections are higher than the DDA ones for small ice particles, but lower for larger particles. These differences may explain

why the soft-spheroid approximation is satisfactory for radar reflectivity simulations, the errors of computed cross-sections for larger and smaller particles compensating each other.

## 1 Introduction

Radar-based quantitative precipitation estimation (QPE) is a challenging task. To derive a relation between radar variables and rainfall rate knowledge of the rain drop size distribution is required. For snowfall, this problem is compounded by the

20 uncertainty in ice particle microphysical and microwave scattering properties. Due to the large variability of snow properties



(size, shape, density and fall velocity), snowfall QPE using radar measurements are more uncertain if compared to rainfall estimation (Matrosov, 1992; Rasmussen et al., 2003; von Lerber et al., 2017).

The relation between equivalent reflectivity factor, $Z_e$, and snowfall intensity, $S$, is usually assumed to follow a power-law form defined by two parameters, i.e. the prefactor $a$ and exponent $b$. These parameters have been derived for weather radars operating in the centimeter wavelength range, either by using observations of radar reflectivity and snowflake size distribution (Gunn and Marshall, 1958; Sekhon and Srivastava, 1970; von Lerber et al., 2017), or by exploiting measurements of radar reflectivity values and coinciding data of snowfall rate (Boucher and Wieler, 1985; Carlson and Marshall, 1972; Fujiyoshi et al., 1990). The $Z_e$-$S$ relationship applicable to mm-wavelength radars were derived in (Matrosov, 2007; Matrosov et al., 2008). These studies have showed that cloud radars at Ka and W band can be used to estimate snowfall accumulation and the vertical structure of snowfall rate (Mitchell, 1988).

Accurate snowfall retrieval algorithms using millimeter wavelengths are of paramount importance considering the increasing number of ongoing and planned satellite cloud and precipitation radar missions. The National Aeronautics and Space Administration (NASA) is currently operating the CloudSat (Stephens et al., 2002) mission carrying the W-band nadir pointing Cloud Profiling Radar (CPR). The NASA/JAXA Global Precipitation Measurement (GPM) core observatory has been launched in 2014 (Skofronick-Jackson et al., 2017) and carries the Dual-frequency (Ku and Ka-band) Precipitation Radar (DPR). Finally, the European/Japanese (ESA/JAXA/NICT) *EarthCARE* mission (Illingworth et al., 2015) is planned to be launched in 2018 and will carry an innovative CPR, which will be the first W-band radar on-board a satellite with Doppler measurement capabilities. Furthermore, a number of ground observatories operate mm-wavelength cloud radars, see for example (Kollias et al., 2007) and (Illingworth et al., 2007).

In Petty and Huang (2010); Botta et al. (2010); Tyynelä et al. (2011) it was argued that for mm-wavelength radars the connection between scattering and microphysical properties of snowflakes is not as straightforward as was previously expected. It was presented that the use of soft-spheroid model, where ice particles are modeled as spheroids with dielectric properties derived from particle density using an effective medium approximation (EMA), may result in a significant underestimation of the radar cross sections. Kneifel et al. (2011) have demonstrated that deviations from the soft-spheroid particle model can be detected in the triple-frequency space, observations of which were reported by Leinonen et al. (2012); Kulie et al. (2014). Kneifel et al. (2015) have shown that the soft-spheroid particle model tend to fail in cases where large low-density aggregates are observed. Given the mounting body of evidence that the relatively simple soft-spheroid models may not be capable of capturing the complexity of ice particle and therefore establish the link between physical and scattering particle properties, the applicability of the $Z_e$-$S$ relationships derived for mm-wavelength radars needs to be reevaluated.

To address this topic, the present study aims to establish and evaluate $Z_e$-$S$ relations at X, Ka and W band by combining the multi-frequency radar measurements and collocated ground observations. The presented dataset is collected during the BAECC measurement campaign that took place at the University of Helsinki research station in Hyytiälä, Finland. Four snowfall cases, comprising of various snowfall regimes and snow microphysical properties, are analyzed. In order to check whether the derived multi-frequency $Z_e$-$S$ relations can be explained by using soft-spheroid particle models, scattering simulation using TMM and DDA were carried out. Observations from from the Particle Imaging Package (PIP) (Newman et al., 2009; Tiira et al.,



2016) were used to constrain these scattering computations. The PIP measures particle size distribution and fall velocities (Tiira et al., 2016). From these observations particle masses were derived (von Lerber et al., 2017) using the hydrodynamic theory (Böhm, 1989). Given particle dimension and mass, corresponding scattering properties were retrieved from a scattering database (Leinonen and Szyrmer, 2015) or equivalent refractive index was computed using Maxwell-Garnett EMA (Sihvola,

1999). From the computed equivalent radar reflectivity factors and measured snowfall rates, $Z_e$-$S$ relations were derived and compared against the previously retrieved radar-based relations.

This paper is organized as follows. The BAECC campaign setup, including an analysis of the calibration and attenuation corrections applied to radar measurements, is given in Section 2. The methodology, used to derive $Z_e$-$S$ relationships from empirical measurements and the details about the single-scattering computations, are described in Section 3. Results from the

field observations and numerical analysis are shown and discussed in Section 4. Section 5 draws final conclusions and remarks.

## 2   Measurements and data

In 2014 the University of Helsinki Hyytiälä Forestry Field Station hosted an 8-months measurement campaign, BAECC (Petäjä et al., 2016). BAECC was jointly organized by the University of Helsinki (UH), the U.S. Department of Energy ARM program, the Finnish Meteorological Institute (FMI) and other international collaborators. During the main campaign, a snowfall

intensive observation period (BAECC SNEX IOP) took place between 1 February - 30 April 2014. It was carried out in collaboration with the NASA GPM ground validation program (Petäjä et al., 2016). BAECC SNEX IOP focused on surface observations of snowfall microphysical properties in combination with multi-frequency radar measurements to establish a link between physical and scattering properties of ice particles. In this study observations of this IOP are used. The surface-based snowfall measurements were carried out by the PIP video-disdrometer (Newman et al., 2009; Tiira et al., 2016) and a weighing

precipitation gauge OTT Pluvio$^2$. The multi-frequency radar observations were obtained by the X-band scanning ARM cloud radar (XSACR), Ka-band ARM zenith radar (KAZR), and the Marine W-band ARM cloud radar (MWACR), which all were part of the second ARM mobile facility (AMF2) deployed at the measurement site during BAECC. In addition to these radars, an FMI operational C-band dual-polarization Doppler weather radar (IKA), located $64 \, \mathrm{km}$ west from Hyytiälä in Ikaalinen, is employed as a reference in the cross-calibration procedure of the ARM radars, as discussed below.

### 2.1   Surface precipitation measurements

The PIP video-disdrometer measures hydrometeor size, fall velocity, shape and particle size distributions (PSD). Here, data are used for characterizing the microphysical properties of the snowfall and for defining the mass-dimensional $m(D)$ relations during the snowfall events. The PIP instrument works in the same way as its predecessor, the Snow Video Imager (SVI) (Newman et al., 2009), but using a camera with a higher frame rate of 380 frames per second. The 2D-gray scale images of falling particle

are obtained, when it falls between the camera and the lamp (distance between the two is $2 \, \mathrm{m}$) and from these multiple images the particle fall velocity is derived. The camera focal plane is at $1.3 \, \mathrm{m}$ and the field of view is $64x48 \, \mathrm{mm}$ with a resolution of





0.01 mm$^2$. For each particle, PIP records the disk-equivalent diameter $D_{Deq}$, which is the diameter of a disk with the same area as the particle shadow.

Particles smaller than 14 pixels (approximately $D_{Deq} < 0.2$ mm) or particles only partly observed or out of focus (blurred) are rejected by the detection software (Newman et al., 2009). Because of the blurring effect, the sizing standard error is

estimated to be 18% (Newman et al., 2009). Also, other shape-descriptive particle parameters are retrieved with the imaging processing software (National Instruments IMAQ) such as particle orientation, total area, and bounding box width and height. Particle fall velocities are recorded as a function of $D_{Deq}$ and values are considered reliable, if there are more than two observations of the identified particle and values are higher or equal to $0.5$ ms$^{-1}$ (PIP software release 1308). In later software versions the fall velocity threshold is removed. The PIP dataset includes PSD in m$^{-3}$mm$^{-1}$ for every minute. It is also

determined as a function of $D_{Deq}$ and subdivided into 105 bins (from 0.125 to 25.875 mm) with the last bin containing particles larger than 25.875 mm.

In this study 5-minute time series of the observed PSD, the fitted $v(D)$ and the retrieved $m(D)$ relations are utilized (Tiira et al., 2016; von Lerber et al., 2017) as a function of the diameter $D$. The PSD is averaged using one minute observations, after spurious particle records are filtered out. The $v(D)$ relation is derived by a linear regression fit in the log-space for the observed

particles during every five minutes (Tiira et al., 2016). The $m(D)$ relation is retrieved by utilizing the general hydrodynamic theory (Böhm, 1989; Mitchell and Heymsfield, 2005), where a snow particle mass is computed from the observed dimension, fall velocity and area ratio of a snow particle. The PIP observes falling particles from the side, whereas the particle dimensions projected to the flow are needed for the hydrodynamic calculations. In von Lerber et al. (2017), errors associated with the observation geometry, and also with the measured PSD were addressed by devising a simple correction procedure; the value of

the correction was chosen for each snow event by comparing the estimated accumulated precipitation to gauge measurements. Similar to $v(D)$ relation, the power-law $m(D) = a_m D^{b_m}$ fit is determined by a linear regression fit in the log-space for the computed particle masses every five minutes.

The weighing precipitation gauge, OTT Pluvio$^2$ 200, records every minute the bucket weight expressed in mm. The gauge is located on a platform at 2 m height surrounded by a double wind fence similar to Double Fence Intercomparison Reference

(DFIR)-fence (Goodison et al., 1998). In addition, the gauge has a Tretyakov wind shield. The Hyytiälä measurement site is surrounded by boreal forest, and therefore, the wind effects are usually moderate. The PIP measurement volume is open and typically affected less by the wind than instruments with enclosed sampling volumes (Nešpor et al., 2000). Therefore, in these wind conditions, the expected wind induced errors are expected to be small.

## 2.2 Ikaalinen C-band weather radar

The dual-polarization Doppler weather radar, used for our analysis, belongs to the Finnish weather radar network (Saltikoff et al., 2010). The radar operates in the C-band and is situated in Ikaalinen (IKA) at circa 64 km west of Hyytiälä. The antenna has an half-power beam widths of $1°$. The radar performs volume scans, repeated every 5 min, and range height indicator (RHI) scans over the Hyytiälä site every 15 min.





The IKA data is quality controlled and calibrated using a number of techniques. The engineering calibration, where different radar components are characterized, is performed during the radar installation and after major system modifications (Saltikoff et al., 2010). In addition to the engineering calibration, the radar receiver and antenna pointing are monitored using sun observations (Huuskonen and Holleman, 2010). The differential reflectivity calibration is monitored using a combination of vertically

pointing scans and sun observations. During the summer months, the IKA radar absolute calibration was checked using the polarimetric self-consistency principle (Gorgucci et al., 1992; Gourley et al., 2009).

Given the continuous monitoring of the radar stability and regular calibration, we use the IKA observations as the calibration standard for the ARM radars. This approach allows us to cross-calibrate the ARM radars even in presence of radome attenuation caused by e.g. large snow accumulation.

## 2.3 ARM cloud radar system calibration at X, Ka and W band

The ARM cloud radar systems operating at X, Ka and W band are integral part of the BAECC snowfall IOP. The antennas of the XSACR, X-band scanning ARM cloud radar, and KAZR, Ka-band ARM zenith radar, are mounted on top of two containers located 17 m away from each other. The MWACR, marine W-band ARM cloud radar, is mounted on the same container as KAZR. All the ARM radars make zenith-pointing observations and the systems have been carefully aligned to

zenith direction during the campaign. Looking at the radar technical properties in Table 1, the range gate spacing and the temporal sampling are comparable, but there is a difference in the beam width between XSACR and the other two systems. To reduce the beam mismatch and to facilitate comparison to the ground-based sensors, all the radar data are averaged to 5-minutes. To derive consistent X-, Ka- and W-band $Z_e$-$S$ relations, the measured radar reflectivity factors were calibrated and corrected for attenuation. The absolute calibration of ARM cloud radars has been performed at the beginning of the BAECC

IOP using engineering calibration and external standard target procedure.

We have also performed a cross-calibration in order to reduce biases between different radar systems. The cross-calibration method is based on the assumption that in the low reflectivity region at the cloud top the small crystals basically scatter in the Rayleigh regime (Hogan et al., 2000). In these regions, therefore, the measured radar reflectivity values from by all millimeter-wave radars should match. As mentioned in Section 2.2, the IKA radar observations are considered to be the reference for

this analysis. The main reason for this selection is that the IKA radar is very stable and its performance is well-monitored. Additionally, given its operating frequency it does not suffer from attenuation during winter storms. Figure 1 (a) shows the profile of 15 February 2014 at 17:13 UTC in which we performed the calibration between 4 and 6 km and in Figure 1 (b) the histograms of the three different calibration errors. The calibration error, measured as the standard deviation of the histograms in Figure 1 (b), shows that the best result is for the error between Ka- and W-band and the worse is for C- and X-band, this

being related mostly to the beam width. Looking at Table 1 larger is the beam width greater is the measured dispersion and vice versa.

One of the reasons for differences in reflectivity measurements can be also attributed to the radome attenuation. For example, the flat shape of the KAZR radome increases the possibility of heavy snow accumulation during a storm. Consequently, when the temperature goes beyond the melting point of ice, the melting snow could produce heavy attenuation that should be





monitored. On the other hand, the conical shape of the MWACR radar limits the amount of accumulated snow, but because of the higher operating frequency is more sensitive to the freezing rain/drizzle. To monitor the radome attenuation sky-noise analysis has been performed for the millimeter-wavelengths radars, KAZR and MWACR. The stability analysis made with the sky noise is shown in Figure 2 as a histogram of sky-noise power measured during ten days of BAECC IOP. The standard

deviations is around $0.25$ and $0.14$ dBm respectively for KAZR and MWACR radar; these values validate the selection of one case during the ten days to operate the cross-calibration. For the Ka-band is visible in Figure 2 a second Gaussian-like with less occurrence centered around $-68.06$ dBm and is due to the previously explained radome attenuation problem.

During the BAECC IOP, radiosondes were launched four time a day. Using these observations as the input to the millimeter-wave propagation model (MPM) (Liebe, 1985), the two-way gaseous path attenuation was computed. This analysis has been

performed for all the dataset. For the 15 February 2014 at 17:24 UTC, the Ka-band attenuation in dB is: $0.1518$ (line-by-line attenuation), $0.0299$ ($O_2$ attenuation), $0.0351$ ($H_2O$ attenuation), whereas the two-way gas attenuation is $0.4335$. For the same time sample, the W-band attenuation in dB is: $0.2228$ (line-by-line attenuation), $0.0288$ ($O_2$ attenuation), $0.2588$ ($H_2O$ attenuation) and the two-way gas attenuation is $1.0206$. As expected, the attenuation for the W-band is twice as large as for Ka-band. By taking into account the gaseous attenuation, the radar calibration offsets during the snowfall experiment were

estimated as $2.9$ dBZ for the XSACR, $3.9$ dBZ for the KAZR and $4$ dBZ for the MWACR.

## 3   Methods

The focus of this study is to investigate the consistency of $Z_e$-$S$ relations at different frequencies, namely at X-, Ka- and W-band. Both the retrieved liquid-water-equivalent snowfall rate $S$ and the equivalent reflectivity factor $Z_e$ are computed from ground observations. Two different computational scattering methods have been applied to link the microphysical properties

of snow particles with the scattering characteristics at all the considered frequencies.

### 3.1   Deriving $Z_e$-$S$ relations at X, Ka and W band

The equivalent reflectivity $Z_e$, measured by the radar systems at different wavelengths, and the liquid-water-equivalent snowfall rate $S$, evaluated from PIP, are the two correlated variables. The liquid-water-equivalent snow rate $S$ (in mm/hr) is derived from the well-known formula (von Lerber et al., 2017)

$$S = \frac{3.6}{\rho_w} \int m(D_{Deq}) v(D_{Deq}) N(D_{Deq}) dD_{Deq}, \tag{1}$$

where $m$ is the mass (in g), $v$ is the velocity (in cm/s), $N$ is the particle size distribution (PSD, in $\text{mm}^{-3}\text{m}^{-1}$) and $\rho_w$ is the liquid water density (in $\text{gcm}^{-3}$). In (1) all quantities are expressed in terms of the disk-equivalent diameter $D_{Deq}$ and derived from PIP measurements (von Lerber et al., 2017).

The radar data used in this study was collected in the vertical pointing mode. To match radar and in-situ measurements, the

radar data collected at the lowest meaningful altitude was used. Given different radar specifications, see Table 1, the Fraunhofer far-field distance for the radars is different. This distance defines the near-field of the radars and is related to the radar antenna



size. The beamwidth difference is related to the antenna diameter that is respectively for XSACR, KAZR and MWACR, 1.82; 1.82; 0.9 m so that the Fraunhofer distance $(2D^2/\lambda)$ is approximately 214 m for XSACR, 773 m for KAZR and 514 m for MWACR. Taking into account the near-field influence, all radar data are selected at 400 m thus guaranteeing to minimize the near-field effects (Sekelsky, 2002).

Another important aspect is related to the different time acquisitions for the various instruments. In Table 1 we note that the temporal sampling of the radars is 2 s whereas for the PIP instrument is 1 min. To avoid the introduction of spurious results, we have decided to average data within a time-window of 5 min. As mentioned in Section 2, the averaging is also useful to tackle the differences in radar beamwidths .

The $Z_e$-$S$ reference model is a power-law form that is $Z_e = aS^b$ where $Z_e$ is expressed in $\mathrm{mm^6 m^{-3}}$ and $S$ is in $\mathrm{mm/h}$
(Carlson and Marshall, 1972; Matrosov et al., 2008). In order to estimate the regression coefficients, we can choose a non-linear least squares in the variable linear space or a linear least squares in the log-log variable space. We have adopted the last approach by applying a linear regression as in (Boucher and Wieler, 1985). The applied log-log model is given by:

$$log_{10} Z_e = b \log_{10} S + \log_{10} a. \qquad (2)$$

where $Z_e$ can be either the time-averaged range-resolved co-polar radar measurement (disregarding the near-field effects) or
15 the numerically simulated backscattering radar response.

### 3.2 Multi-frequency $Z_e$-$S$ relations using T-matrix scattering model

Single scattering computations are performed using the TMM code by Mishchenko (Mishchenko, 2000) in the version developed by Leinonen (Leinonen, 2014). The purpose is to obtain a good approximation for the equivalent reflectivity $Z_e$ in order to investigate the $Z_e$-$S$ relations in snowfall from the in-situ microphysical parameters. For what concerns the snowflake shape,
the TMM allows to model the nonspherical hydrometeors as spheroids. The radar community usually models raindrops using a spheroidal model and, to some extent the latter is also useful for more complicate particle structures such as snowflakes (Matrosov, 2007). This is because the microwave backscatter properties do not depend on the small details, but mostly on the overall structure, at least at cm-wavelengths.

The most important parameter to characterize the whole structure is the spheroid aspect ratio $r_s = b_s/a_s$ where $a_s$ and $b_s$
are the horizontal and vertical dimensions of the spheroid ($r_s = 1$ spherical particle, $r_s \geq 1$ prolate particle and $r_s \leq 1$ oblate particle) (Dungey and Bohren, 1993). The spheroidal model is useful for smaller snowflake size but it looses its applicability for very high frequency, such as W band, in case of larger snowflakes with very small aspect ratio (dendrities) as seen in (Tyynelä et al., 2011). The spheroidal model can be considered complete if the average in time weakens the effect of the particle complex structure usually visible only for short time period of observations (Leinonen, 2014). Using the spheroid aspect ratio as a tuning
parameter, two different behaviors can be identified: $r_s$ remains approximately constant changing the frequencies for larger particle (for example in case of rimed particles) and $r_s$ changes from larges to smalls values increasing the frequencies for smaller particle (for example in case of low-density fluffy particles) (Magono and Nakamura, 1965; Matrosov, 2007; Leinonen, 2014).





The snowflakes, due to aerodynamic forcing, typically fall with the major axis oriented horizontally (Magono and Naka-mura, 1965; Matrosov, 2007). Following a convention presented in (Matrosov, 2007; Matrosov et al., 2008), the TMM is set with single shaped snowflakes oriented preferably horizontally with the $0°$ and standard deviation of $10°$ with respect to the vertical. As mentioned in Section 2.3, all radars at X-, Ka- and W-band are vertically pointing system so that there is no polar-

ization dependence of reflectivity measurements. However, the latter may be influenced by the different instrumental standard deviations of their receiving systems.

To better define the influence of the snow aspect-ratio tuning parameter, we can define the density $\rho$ (in $g/cm^3$) as dependent from it:

$$\rho = \frac{m}{\pi/6 D_{Veq}^3} \tag{3}$$

in which the mass $m$ is defined as in (von Lerber et al., 2017) and $D_{Veq}$ is the diameter volume equivalent defined from $D_{max}$, the maximum diameter obtained by PIP (von Lerber et al., 2017), as $D_{Veq} = r_s^{1/3} D_{max}$. The presence of the aspect ratio inside the density reflects its dependency on the complex refractive index of snow $m_S$ that is defined through the Maxwell Garnett effective medium approximation (EMA).

The $\Gamma$-size distribution (in $mm^{-1}m^{-3}$) is assumed to model the snowflake number concentration:

$$N(D_{Veq}) = N_{w,Veq} f(\mu_{Veq}) \frac{D_{Veq}}{D_{0,Veq}}^{\mu_{Veq}} exp(-\Lambda_{Veq} D_{Veq}) \tag{4}$$

where $N_{w,Veq}$ is the intercept parameter (in $mm^{-1}m^{-3}$), $f(\mu_{Veq})$ and $\mu_{Veq}$ parameter are dimensionless, and $\Lambda_{Veq}$ is the slope of the distribution in $1/mm$. This $\Gamma$-size distribution can be expressed starting from the moments of the snowflakes distributions measured by PIP, as in (Bringi and Chandrasekar, 2001), taking into account the variable changing from $D_{max}$ to $D_{Veq}$ as follows

$$N_{w,Veq} = N_{w,max} \frac{dD_{max}}{dD_{Veq}} = N_{w,max} \frac{1}{r_s^{1/3}} \tag{5}$$

with

$$D_{0,Veq} = D_{0,max} r_s^{1/3} \tag{6}$$

$$\Lambda_{Veq} = \Lambda_{max} \frac{dD_{max}}{dD_{Veq}} = \Lambda_{max} \frac{1}{r_s^{1/3}} \tag{7}$$

$$\mu_{Veq} = \mu_{max}. \tag{8}$$

The equivalent reflectivity $Z_e$ of a snowflake ensemble is obtained by integrating the TMM-derived radar cross section of a single particle with the $\Gamma$-size distribution in equation 4.



### 3.3 Multi-frequency $Z_e$-$S$ relations using DDA scattering model

The DDA model is used to characterize the single scattering properties of snowflakes using complex and realistic shape models of snow particles. Although the PIP instrument records a 2D image of the falling particles DDA is not used to compute the single scattering properties of individual observed snowflakes, but rather it is used to compile lookup tables (LUT) of scattering properties of generic realistically shaped particles. In fact, even though sophisticated particle imager can provide detailed description of the particle morphology by means of a multi-angle view (Garrett and Yuter, 2014) the internal structure of snowflakes is still not observable; moreover the huge computational cost of DDA would make such approach prohibitive using current available calculators.

LUT can be used to estimate the average scattering properties of observed ice particles provided some constrains on the microphysical properties of snow. In particular Leinonen and Szyrmer (2015) have published an extensive LUT of backscattering properties for realistically shaped snow particle models. The shape model is obtained by accurately simulating the microphysical processes that lead to snowflake growth through their falling path. In particular, the snowflake formation is simulated by aggregation of pristine dendrites and subsequent or simultaneous riming of those aggregates using multiple values of equivalent liquid water path which in turns drive the riming degree. The simulation of the riming process allows the scattering database to span through a large range of particle masses and sizes allowing to use those microphysical features to constrain the ice particle scattering properties. Moisseev et al. (2017) have shown that during BAECC experiment snow particle were moderately to heavily rimed, therefore the selection of the database that includes rimed particles is warranted. The scattering properties of the simulated particles are in fact picked from the LUT by finding the entries that most closely match the particles size and mass as it has been retrieved in (von Lerber et al., 2017) as it follows.

For each size range defined by the PSD discretization operated by PIP the LUT is filtered for the particles with a size that falls within the PSD size bin. Than, using the retrieved m-D relation the LUT entries are sorted with respect to the difference between their mass and the expected particle mass computed using the retrieved mass-dimensional relation. An arbitrary number of 10 entries that most closely match the retrieved m-D relation are selected and their scattering properties are averaged in order to define the representative backscattering cross section of that particular size range. Larger number of particles can be picked from the LUT in order to represent a larger variability of particle mass, but the effects of including heavier and lighter particles tends to cancel out in the averaging and does not produce appreciable differences in the final integrated reflectivity.

It is worth noting that scattering properties are computed for randomly oriented particles at fixed orientation (Leinonen and Szyrmer, 2015), meaning that each particle composing the database does not have any preferential orientation, but its scattering properties are simulated at a fixed orientation. The inclusion of multiple particles within each size range is then required to take into account also for the orientation averaging rather then just the inclusion of some variability in the microphysical properties of realistic snowflakes.



## 4   Results

The results are shown for four BAECC case studies to investigate the consistency of $Z_e$-$S$ relations at X-, Ka- and W-band using surface observations. Indeed, ten case studies are available from the BAECC IOP, but only for the selected four events millimeter-wave radars (Ka- and W-band) together with the PIP instrument can be considered well calibrated, in the other

cases effects of the random attenuation cannot be fully compensated. The uncertainties of the $Z_e$-$S$ parametric relations at different frequencies are also investigated using TMM and DDA numerical results. TMM and DDA results can provide some microphysical insights of the considered snowfall events.

### 4.1   Analysis of X, Ka and W band $Z_e$-$S$ empirical relations

The consistency of $Z_e$-$S$ relations at different frequencies (X-, Ka-, W-band) is highlighted in the following Figures 3, 4, 5

and 6. The avialable BAECC dataset has been divided into 2 main sub-datasets: i) one having a smaller median diameter $D_0$ mainly characterized by fluffy snow (case study on 12 February 2014); ii) the other one with the remaining data with higher median diameters (black points), dominated by rimed snow (all other case studies on 15/16, 21/22 February and 20 March 2014). The selection has been carried out with a threshold on the median diameter $D_0$, as it will be discussed later on.

The case study of the 12 February 2014 between 04:00 and 08:50, characterized by fluffy snow, is shown in Figure 3. The

latter shows the retrieved liquid-water-equivalent snowfall rate $S$ from PIP (see in equation (1)) with respect to the measured equivalent reflectivity factor $Z_e$ from the ARM radars. We have adopted a representation with $Z_e$ in dBZ and $S$ in base 10 logarithm to adhere to the log-log model in equation (2). The two sub-datasets, highlighted by light grey and black points, are representative of the 2 different snow types. The regression analysis has been applied only to the smaller-diameter data (black points) because the others are statistically few points.

Three $Z_e$-S parametric models (black dashed lines) at X-, Ka- and W-band are also shown and the regressive coefficients of the power-law formula are reported in Table 2. The accuracy of $Z_e$-$S$ relations has been evaluated using the root mean square error (RMSE) (in $\mathrm{dBZ}$) (where the error is defined as the difference between $Z_e$ estimated from PIP and observed from the ARM radars). To facilitate the result readability, the normalized RMSE (NRMSE) has been also introduced as the error standard deviation normalized to the value range (maximum minus minimum value). The percentage NRMSE of the regression

algorithms, shown in Figure 3, is about 13%, 15% and 21% respectively for X-, Ka- and W-band. The regression coefficients have a trend different from that discussed in (Matrosov, 2007; Matrosov et al., 2008). The a-coefficients are quiet similar at X-/Ka-band and slightly different at W-band. The b-coefficients show a small variation, but they increase with the frequency. This behavior can be attributed to data from the 12 February 2014 case where fluffy snowflakes were prevailing. On the basis of these last remarks, Table 2 also summarizes the findings $Z_e$-$S$ relationships for the 2 main snowfall regimes, fluffy and

rimed snowfall. These general relationships can be derived by putting together data from the corresponding case studies, 12 February 2014 and 15/16 Feb., 21/22 Feb. and 20 March 2014, respectively. The Table 2 shows the empirically-derived a and b regressive coefficients at X, Ka and W for both fluffy and rimed snowfall with their mininum-maximum variability.



Figure 4, 5 and 6 show three case studies where a rimed snowfall process is supposed to be predominat, similarly to what shown in  Matrosov (2007); Matrosov et al. (2008). As in Figure 3, we measured $Z_e$ versus $S$ from PIP at the various frequency bands. Figure 4 and 5 depicts data from the 15/16 February 2014 case between 21:00 and 01:48 UTC and from the 21/22 February 2014 case between 16:00 and 03:24 UTC, respectively. All data (black points) have been taken into account for the data regression (dashed black lines). Looking at Table 2, these two cases have uniform a-coefficients with values around 61, 35 and 7 for X, Ka and W band (similar to the value in  Matrosov (2007); Matrosov et al. (2008)). The b-coefficients are also very similar for the X-band at around 1.2 and a little bit different for the Ka- and W-band but maintaining the trend that increasing the frequency the b-coefficient decreases. The accuracy expressed in terms of percentage NRMSE is also relatively low with values around 17% (X-band and Ka-band) and 15% (W-band) for the 15/16 February and a little bit worse around 19% (X-band and Ka-band) and 16% (W-band) for 21/22 February.

The last case study of 20 March 2014 between 16:00 and 20:48 UTC, dominated by rimed snowfall, is presented in Figure 6. Only the KAZR and MWACR are available for the dataset because of the XSACR radar was non-activated. The values of the a- and b-coefficients are similar to the previous two cases and also with Matrosov (2007); Matrosov et al. (2008) having a NRMSE accuracy of about 18% (Ka band) and 21% (W band).

The correlation between the predicted and measured $Z_e$ can provide a visual appreciation of the proposed parametric model accuracy. Figure 7 shows measured $Z_e$ from the ARM radars as a function of $Z_e$ estimated from the power-law form $Z_e = aS^b$ with the coefficients of Table 2. This plot confirms the consistency of the $Z_e$-$S$ relation at X, Ka and W band, whereas the difference between the 12 February 2014 and the last three cases is also apparent thus highlighting the impact of snowfall type. Note that, for the fluffy snow of the 12 February 2014 case, reflectivity data show the same dynamic range for all radar frequency bands. On the contrary, the other rimed-snow case studies mark a cleare separation of the reflectivity dynamic range at the different frequency bands. By merging all 3 case study data, the overall percentage NRMSE of $Z_e$-$S$ relations in the considered cases are within 13%-19% at X-band, 15%-19% at Ka-band and 15%-21% at W band.

Using all collected BAECC measured data, the frequency behavior of the $a$ and $b$ power-law coefficients in Table 2 may be useful to suggest a general trend of the $Z_e$-S relation, even though only 3 frequency samples at X, Ka and W band are available. Figure 8 shows the spectral variation of the $a$ and $b$ coefficients, splitting the results between fluffy snow (blue points) and rimed snow (red points). The confidence interval at 95%, represented by transparent blue for fluffy snow and transparent red for rimed snow and derived from the conventional statistical t-test, is also shown to quantify the coefficient estimate variability. The spline interpolation has been introduced for both fluffy snow (blue line) and for rimed snow (red line) data to outline a possible general law for these 2 coefficients. Within these limits, it is worth noting: i) the opposite spectral trend of the $b$ coefficient for the 2 snow types in Figure 8 (b); ii) the monotonic decrease of the $a$ coefficient with the frequency for the rimed snow and a parabolic trend for the fluffy snow from Figure 8 (a).

## 4.2   Explaining $Z_e$-$S$ relations with scattering simulations

The temporal trend of multi-frequency radar measurements can provide a further insight into the analysis of snowfall regime and the capability to simulate its behavior. Figure 9 shows the equivalent reflectivity factor $Z_e$ as a function of time for the



fluffy-snow case study of 12 February 2014. The black points correspond to ARM-radar mean $Z_e$, whereas the bars are related to the variation between their minimum and maximum values within the same averaging time interval of 5 minutes. A portion of the measurements (grey points in Figure 9(a)-(d)) at the beginning of the event is disregarded since the variation index (defined as the ratio between minimum-maximum variability interval and its mean value) is considered too high. The different colored

lines refer to $Z_e$, simulated from TMM using PIP data with a variable aspect ratio $r_s$ between 0.2 and 1 with step 0.2. The smaller value $r_s$=0.2 (red line) indicate largely oblate particles, whereas $r_s$=1.0 (blue line) indicate spherical snow particles. By comparing ARM measurements and TMM simulations, the optimal aspect ratio tuning parameter seems to decrease when increasing the frequency: X-band data are better represented by TMM-derived spherical particles ($r_s = 1$), whereas Ka- and W-band results are in agreement with an aspect ratio of $0.6$. After 07:00 UTC within the heavy precipitation period, no data

are available for X-band radar in this case study, but the optimal aspect ratio tend to change to a value around of $r_s = 0.8$ for the millimeter-wave radars (Ka- and W-band).

Figure 10 shows again $Z_e$ as a function of time for the 15-16 February 2014, the rimed snowfall case. We can distinguish two main intervals: before the heavy precipitation around 22:10 UTC, as for the previous case, the optimal aspect ratio decreases when increasing the radar frequency, whereas during the heavy precipitation interval (from 22:50 on) the optimal aspect ratio

seems to be around $0.6$ independently from the frequency. Figure 11 shows the time behavior for the 21-22 February 2014, a rimed-snow case similar to the previous one. Till 22:00 UTC the optimal aspect ratios seem to be around 1, $0.8$ and $0.8$ at X-, Ka- and W-band, respectively, whereas during the heavy precipitation period it is constant around $r_s = 0.6$ irrespective of the frequency. These considerations are also valid for the 20 March 2014 in Figure 12 in which the optimal aspect ratio is about $r_s = 0.6$ for the millimeter-wave radars (Ka- and W-band). For the this case X-band data are not available and thus they are

not shown in the figure.

As a general comment on Figures 9-12, we note that measured data are included inside data uncertainty bounds. The incremental difference in terms of $Z_e$ due to an increase of 0.2 in the particle aspect ratio is about $1.7$ dBZ at X band, $2.5$ dBZ at Ka band and $6$ dBZ at W band. The difference between the value for $r_s = 0.2$ and $r_s = 1$ is on an average equals to $5.5$ dBZ for X-band, $7$ dBZ for Ka-band and $12$ dBZ for W-band. By increasing the frequency from X- to W-band, the radar reflectivity

seems to be, in general, more sensitive to the non-spherical shape of the snowflakes with $r_s$ going from 1 down to 0.6.

TMM numerical simulations can be used to derive a synthetic parametric regressive model, tuned to $r_s$ and to be quantitatively compared with the empirical algorithm given in Table 2. Using the power-law formula, defined in Section 3.1 through the equation 2, Table 3 shows the different $Z_e$-$S$ synthetic relationships obtained as a function of the aspect ratio. From Table 3 we can retrieve the best aspect ratio for each frequency band using by minimizing the RMSE and NRMSE of the synthetic $Z_e$-$S$

relation with respect to the empirical one. The output of this exercise is highlighted in bold black in Table 3. As qualitatively shown by Figures 9-12, for the 12 February 2014 case the optimal aspect ratio is not constant and its value decreases with the frequency being 1 for X-band, $0.8$ for Ka-band and $0.6$ for W-band. For the remaining cases, the optimal aspect ratio seems to be frequency-independent and equals to $0.6$ for 15/16 February and 20 March, and $0.8$ for the 21/22 February. These two different behaviors are related to the presence of fluffy snow for the 12 February 2014 and predominance of rimed snow for

15-16 and 21-22 February 2014 and the whole case of 20 March 2014.



The TMM scattering model is finally compared with the DDA one in order to better evaluate its accuracy for particle scattering simulations, assuming that DDA is better representation of snowflake complex-shape response. Figure 13 shows the W-band co-polar backscatter cross section $\sigma$ of oblate particles as a function of the disk-equivalent diameter $D_{Deq}$ for the 3 rimed-snow case studies. We show only results at W band, the highest available frequency band, since at W band the scattering

regime moves from the Rayleigh to the Mie one and is by far the most sensitive numerical scenario. TMM simulations are in red, green and blue lines referring to different aspect ratios ($r_s = 0.2$, $0.6$, $1$, respectively), whereas DDA method are given by the black line. The dotted line shows the product between the snowflake PSD and the TMM-based backscatter cross section $\sigma$ with aspect ratio of $0.6$ (green) and for the DDA (black) useful to understand which particle size contributes to the whole equivalent reflectivity. This Figure 13 suggests two main remarks: i) DDA results are within the variability of TMM ones

for a variable aspect ratio $r_s$ so that the latter can be considered as a tuning parameter; ii) looking at the product between the PIP-derived PSD and the radar cross section $\sigma$, we note that the TMM-based product is higher than the DDA one for small ice particles and is lower for the larger particles. The latter consideration leads to the conclusion that the soft-spheroid approximation may work rather well for computing radar reflectivity since the errors for larger particles are compensated by those for smaller particles. At the end TMM may be a good compromise between flexibility and accuracy and is able to provide

general $Z_e$-$S$ relations as a function of the aspect ratio (see Table 3).

## 5   Conclusions

The multifrequency $Z_e$-$S$ relationships at X-, Ka- and W-band have been investigated in this work using a high quality dataset of zenith-pointing radar data and in-situ measurements acquired during the BAECC campaign.

From a data analysis point of view, adopting as a reference a power-law relation, regression coefficients have been extracted

for characterizing $Z_e$-$S$ at the considered frequency bands. These coefficients are in line with those provided in literature and confirm also the applicability of a power-law empirical model to the millimeter-wave radars for snowfall estimation in 2 main precipitation modes. The latter can be schematically refer to as rimed snowfall and fluffy snowfall, even though the second one is not fully represented in the selected dataset.

For validation and intercomparison, numerical simulations have been also carried out using the TMM for oblate particles,

coupled with microphysical sizing from an in-situ video-disdrometer and a new mass-dimensional relation and using the particle aspect-ratio as a tuning parameter. Uncertainty of each derived relationship has been provided and ranked with respect to the BAECC radar available measurements. The latter show that there are privileged aspect ratios for the 2 identified rimed and fluffy snowfall regimes. TMM numerical results have been also compared with DDA scattering simulation in order to better understand the role of the aspect-ratio tuning parameter.

Uncertainty evaluation has been attached to each empirical and synthetic power-law relationship at X, Ka and W band for each case study and for the 2 main regimes, fluffy and rimed snowfall. This set of regression coefficients may be used in the future for selecting optimal $Z_e$-$S$ algorithms in different geographical regions and to assess the dependence on the snowfall type. In this respect, the results of this work can represent a first step towards the design of snowfall retrieval algorithm de-



rived from ground-based measurements and the set-up of simplified scattering simulations for radar centimeter and millimeter wavelength.

*Acknowledgements.* MTF has been partly supported by the Center of Excellence CETEMPS, L'Aquila (Italy). The activity of the author DO was funded by the German Research Foundation (DFG) as part of the Emmy-Noether Group OPTIMIce under grant KN 1112/2-1. The author AvL was supported by SESAR Joint Undertaking Horizon 2020 grant agreement No 699221 (PNOWWA). The research of DM was supported by Academy of Finland (grant 305175) and the Academy of Finland Finnish Center of Excellence program (grant 3073314). DM also acknowledges the funding received by ERA-PLANET, trans-national project iCUPE (Grant Agreement n. 689443), funded under the EU Horizon 2020 Framework Programme.



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



**Table 1.** Radar technical specifications are shown for C-band polarimetric Doppler weather radar and for the ARM cloud radar systems at X-, Ka- and W-band.

| Acronym | IKA | XSACR | KAZR | MWACR |
|---|---|---|---|---|
| Location | Ikaalinen | Hyytiälä | Hyytiälä | Hyytiälä |
| Frequency (GHz) | 5.6 | 9.7 | 35.3 | 95.0 |
| Beam width (°) | 0.94–0.98 | 1.27 | 0.33 | 0.38 |
| Sensitivity at 1 km (dBZ) | -48 | -30[a] | -50[a] | -50[a] |
| Range gate spacing (m) | - | 25 | 25 | 30 |
| Temporal sampling | 5 min | 2 s | 2 s | 2 s |

[a] Sensitivity for 2 s integration time and for nominal ARM radar settings.





**Table 2.** Empirical $Z_e$-$S$ for each BAECC case study and for both rimed and fluffy snowfall regimes. $Z_e$-$S$ at X-, Ka- and W-band derived from ARM radar and PIP video-disdrometer using a least-square regressive analysis in the log-log space for each case study. The root-mean-square error (RMSE) is also shown as well as the Normalized RMSE (NRMSE) using for normalization the value range (defined as the maximum value minus the minimum value) of the measured data as normalization. By properly grouping BAECC events, the last 6 rows report $Z_e$-$S$ at X, Ka and W band for the 2 observed snowfall regimes, providing also a and b coefficients variability. A 95% confidence interval for each coefficient, $\Delta_a$ and $\Delta_b$, represents a closed interval where a certain percentage of the population is likely to lie.

| Date | Band | a | b | RMSE dBZ | NRMSE (range) adim. |
|---|---|---|---|---|---|
| 12 Feb. 2014 (Fluffy snowfall) | X | 19.54 | 1.09 | 1.58 | 0.13 |
|  | Ka | 21.67 | 1.17 | 1.90 | 0.15 |
|  | W | 12.46 | 1.25 | 2.99 | 0.21 |
| 15/16 Feb. 2014 (Rimed snowfall) | X | 61.48 | 1.22 | 4.29 | 0.17 |
|  | Ka | 38.39 | 1.06 | 3.74 | 0.17 |
|  | W | 7.65 | 0.73 | 3.26 | 0.15 |
| 21/22 Feb. 2014 (Rimed snowfall) | X | 61.73 | 1.19 | 4.41 | 0.19 |
|  | Ka | 35.23 | 0.83 | 3.24 | 0.19 |
|  | W | 7.00 | 0.63 | 2.44 | 0.16 |
| 20 March 2014 (Rimed snowfall) | Ka | 41.99 | 0.81 | 1.70 | 0.18 |
|  | W | 10.60 | 0.51 | 1.73 | 0.21 |

| Regime | Band | a | b | $\Delta_a$ dBZ | $\Delta_b$ adim. |
|---|---|---|---|---|---|
| BAECC Rimed snowfall | X | 61.71 | 1.20 | 60.45-62.97 | 0.97-1.42 |
|  | Ka | 38.45 | 0.93 | 37.33-39.57 | 0.81-1.05 |
|  | W | 8.35 | 0.74 | 7.25-9.45 | 0.63-0.84 |
| BAECC Fluffy snowfall | X | 19.54 | 1.09 | 17.83-21.25 | 0.71-1.48 |
|  | Ka | 21.67 | 1.17 | 19.76-23.57 | 0.71-1.63 |
|  | W | 12.46 | 1.25 | 9.70-15.22 | 0.52-1.98 |

Four BAECC cases of snowfall events with the a-b coefficients estimated in a 5-minute time window.


**Table 3.** For all BAECC 4 case studies $Z_e$-$S$ (with $Z_e$ in mm$^6$m$^{-3}$, $S$ in mmh$^{-1}$) relationships, derived from TMM-based numerical simulations of $Z_e$ and PIP-derived $S$ and using the oblate-particle aspect ratio as a tuning parameter between 0.2 and 1. The best $Z_e$-$S$ relation is highlighted in bold and corresponds to the power-law minimizing both RMSE and NRMSE.

| Date | Band | $Z_e$-$S$ ($r_s$=0.2) | RMSE dBZ | NRMSE adim. | $Z_e$-$S$ ($r_s$=0.4) | RMSE dBZ | NRMSE adim. | $Z_e$-$S$ ($r_s$=0.6) | RMSE dBZ | NRMSE adim. | $Z_e$-$S$ ($r_s$=0.8) | RMSE dBZ | NRMSE adim. | $Z_e$-$S$ ($r_s$=1) | RMSE dBZ | NRMSE adim. |
|---|---|---|---|---|---|---|---|---|---|---|---|---|---|---|---|---|
| 12 Feb. 2014 Fluffy snowfall | X | $57.44S^{0.88}$ | 6.16 | 0.52 | $31.46S^{0.88}$ | 3.68 | 0.31 | $23.47S^{0.90}$ | 2.53 | 0.21 | $18.93S^{0.91}$ | 1.87 | 0.16 | $\mathbf{15.80S^{0.9}}$ | **1.62** | **0.14** |
| | Ka | $63.34S^{0.83}$ | 6.93 | 0.56 | $32.08S^{0.84}$ | 4.12 | 0.33 | $22.12S^{0.89}$ | 2.60 | 0.21 | $\mathbf{16.48S^{0.95}}$ | **1.94** | **0.16** | $12.71S^{1.02}$ | 2.38 | 0.19 |
| | W | $58.72S^{1.01}$ | 8.66 | 0.61 | $22.90S^{1.12}$ | 4.55 | 0.32 | $\mathbf{11.44S^{1.22}}$ | **2.99** | **0.21** | $5.77S^{1.27}$ | 4.58 | 0.32 | $2.91S^{1.30}$ | 7.23 | 0.51 |
| 15/16 Feb. 2014 Rimed snowfall | X | $195.24S^{1.38}$ | 6.38 | 0.25 | $105.17S^{1.39}$ | 4.76 | 0.18 | $\mathbf{74.13S^{1.38}}$ | **4.36** | **0.17** | $57.17S^{1.37}$ | 4.38 | 0.17 | $46.09S^{1.36}$ | 4.60 | 0.18 |
| | Ka | $181.23S^{1.33}$ | 7.29 | 0.32 | $82.49S^{1.29}$ | 4.78 | 0.21 | $\mathbf{48.18S^{1.23}}$ | **3.85** | **0.17** | $30.29S^{1.16}$ | 3.95 | 0.18 | $19.78S^{1.10}$ | 4.76 | 0.21 |
| | W | $103.23S^{1.10}$ | 11.09 | 0.52 | $28.44S^{0.97}$ | 6.19 | 0.29 | $10.76S^{0.88}$ | 3.51 | 0.29 | $\mathbf{4.71S^{0.83}}$ | 4.02 | **0.16** | $2.29S^{0.79}$ | 6.29 | 0.29 |
| 21/22 Feb. 2014 Rimed snowfall | X | $127.48S^{1.42}$ | 5.17 | 0.23 | $107.41S^{1.30}$ | 4.91 | 0.22 | $82.64S^{1.29}$ | 4.54 | 0.20 | $\mathbf{68.49S^{1.28}}$ | **4.43** | **0.19** | $58.60S^{1.26}$ | 4.44 | 0.20 |
| | Ka | $131.92S^{1.20}$ | 5.76 | 0.34 | $93.20S^{1.10}$ | 4.78 | 0.28 | $60.49S^{1.06}$ | 3.72 | 0.22 | $\mathbf{41.85S^{1.03}}$ | **3.32** | **0.20** | $29.72S^{0.99}$ | 3.52 | 0.21 |
| | W | $73.82S^{1.08}$ | 9.28 | 0.60 | $36.22S^{0.92}$ | 6.77 | 0.43 | $16.05S^{0.88}$ | 3.84 | 0.25 | $\mathbf{7.73S^{0.86}}$ | **2.59** | **0.17** | $3.97S^{0.85}$ | 4.07 | 0.26 |
| 20 March 2014 Rimed snowfall | Ka | $120.18S^{1.11}$ | 5.07 | 0.55 | $55.92S^{1.09}$ | 2.27 | 0.25 | $\mathbf{33.72S^{1.06}}$ | **1.95** | **0.21** | $21.84S^{1.05}$ | 3.22 | 0.35 | $14.68S^{1.03}$ | 4.77 | 0.52 |
| | W | $72.35S^{1.06}$ | 8.90 | 1.06 | $21.41S^{1.03}$ | 3.91 | 0.47 | $\mathbf{8.52S^{1}}$ | **2.10** | **0.25** | $3.83S^{0.99}$ | 4.59 | 0.55 | $1.88S^{1.01}$ | 7.50 | 0.90 |





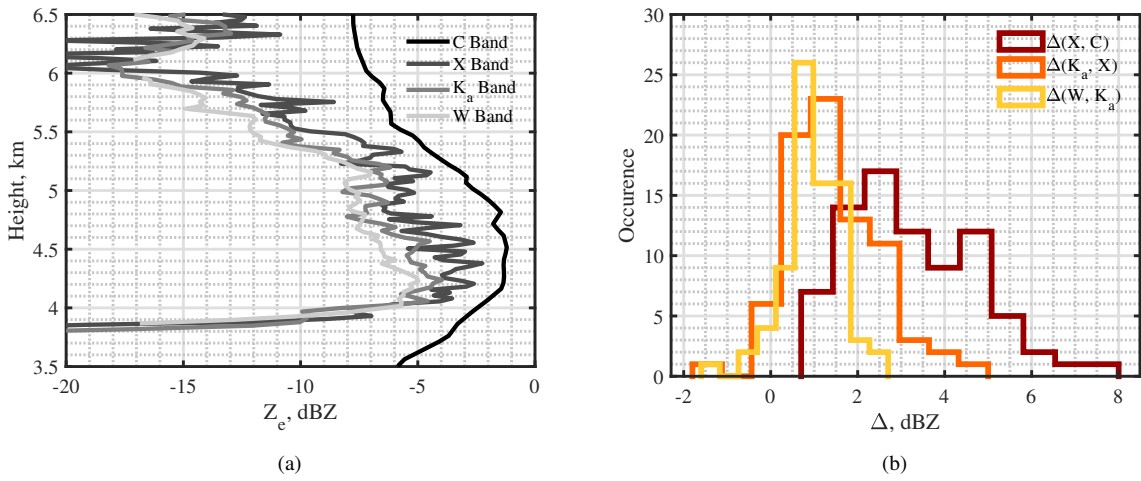

(a)          (b)

**Figure 1.** Left panel shows radar profiles at C-, X-, Ka- and W-band for the 15 February 2014 at 17:13 UTC where calibration is performed within in the most stable height interval between 4 and 6 km. Right panel shows calibration error histograms related to the differences ($\Delta$) between X- and C- band radars (dark red), Ka- and X-band radars (orange), and W- and Ka-radars (yellow).





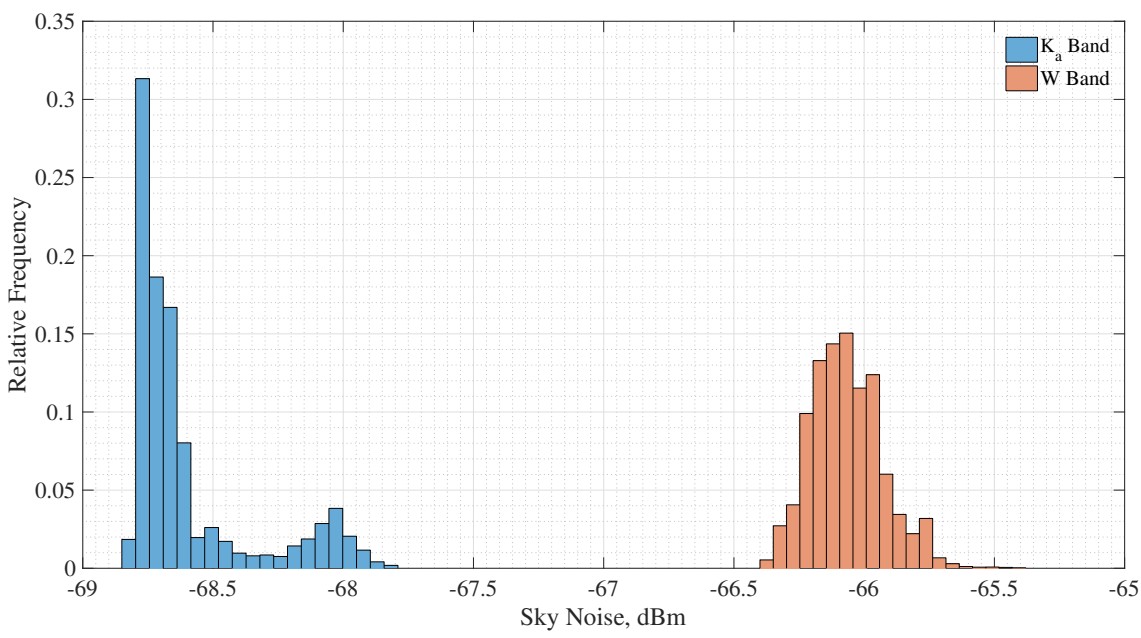

**Figure 2.** Relative frequency histograms of the sky noise antenna temperature for the Ka- and W-band radars for all 10 days of BAECC IOP campaign.





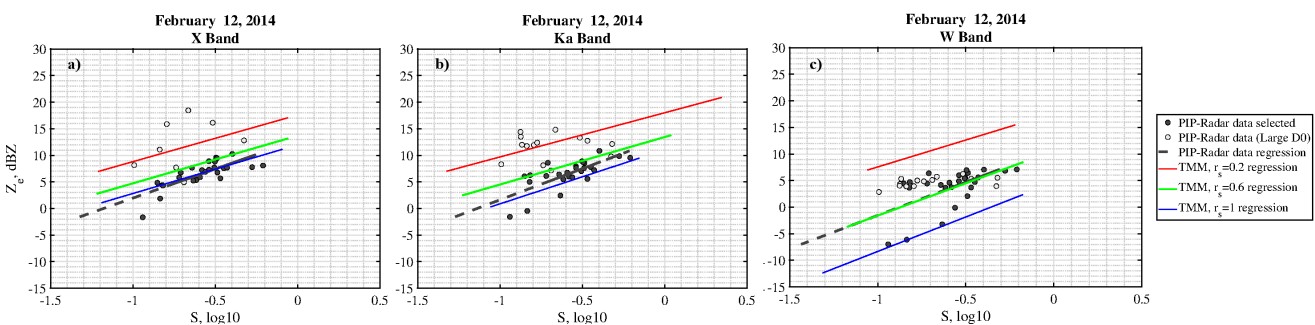

**Figure 3.** Case of study: 12 February 2014 (Fluffy snowfall) - Scatter plot of the equivalent radar reflectivity, measured by ARM radars (black points), with respect to the snow rates S, measured by PIP. The black line represents the $Z_e$-$S$ empirical least-square relationship. $Z_e$-$S$ parametric relations, derived from TMM-based simulations, are also shown for different aspect ratios (0.2, 0.6, 1) using red, green and blue lines.





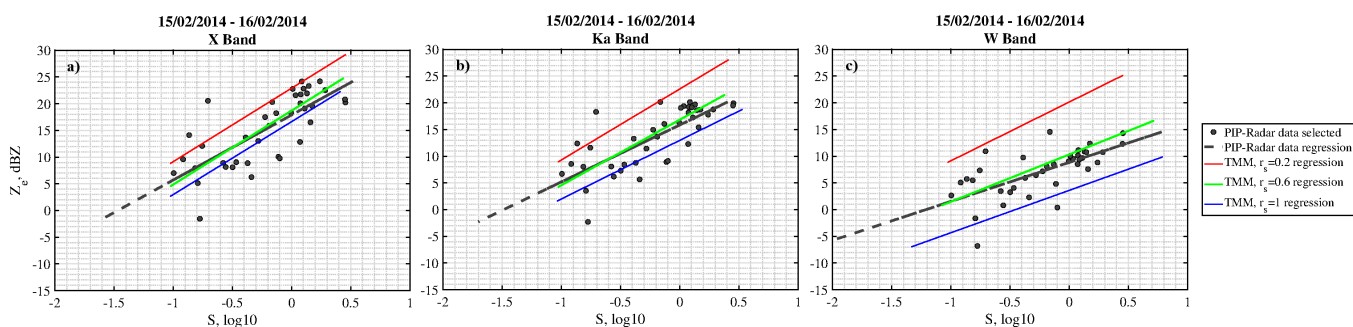

**Figure 4.** Same as Figure 3, but for 15/16 February 2014, 21:00-01:48 UTC (Rimed snowfall).





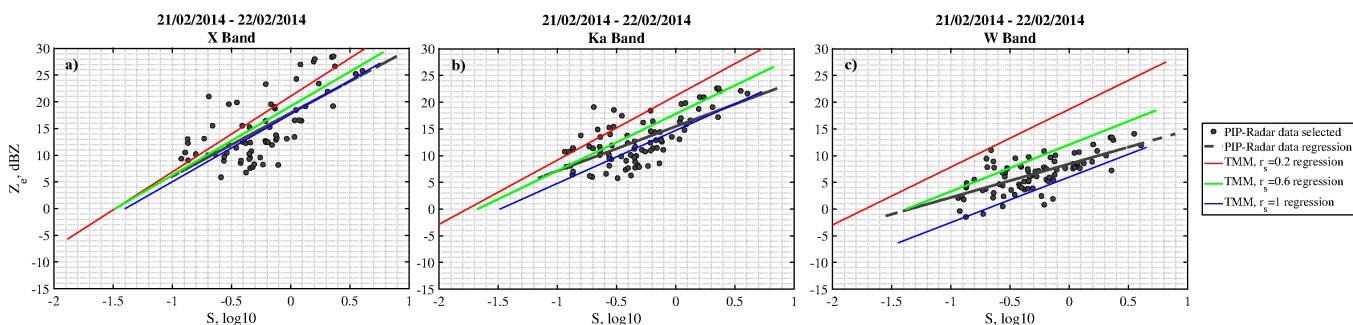

**Figure 5.** Same as Figure 3, but for 21/22 February 2014, 16:00-03:24 UTC (Rimed snowfall).



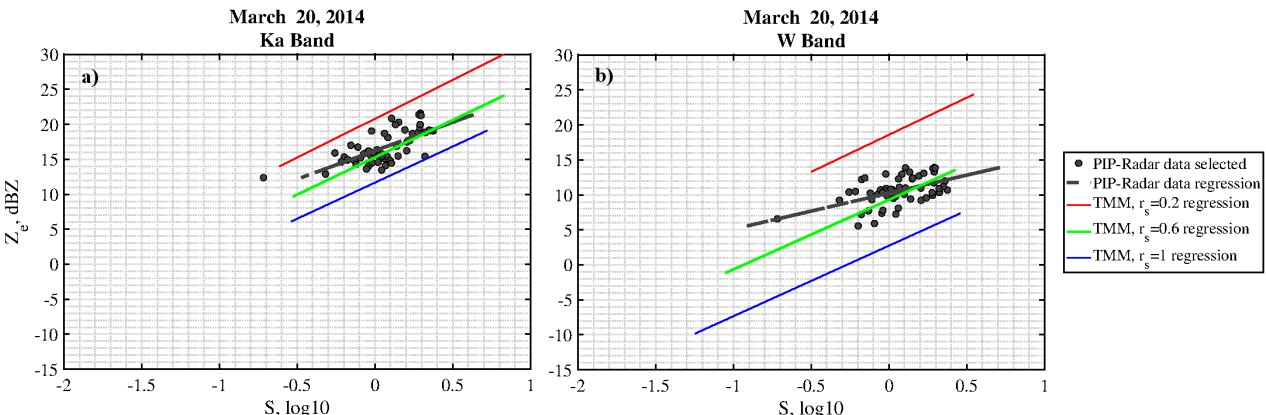

**Figure 6.** Same as Figure 3, but for 20 March 2014, 16:00- 20:48 UTC (Rimed snowfall). The X band radar data are not available for this time-window.



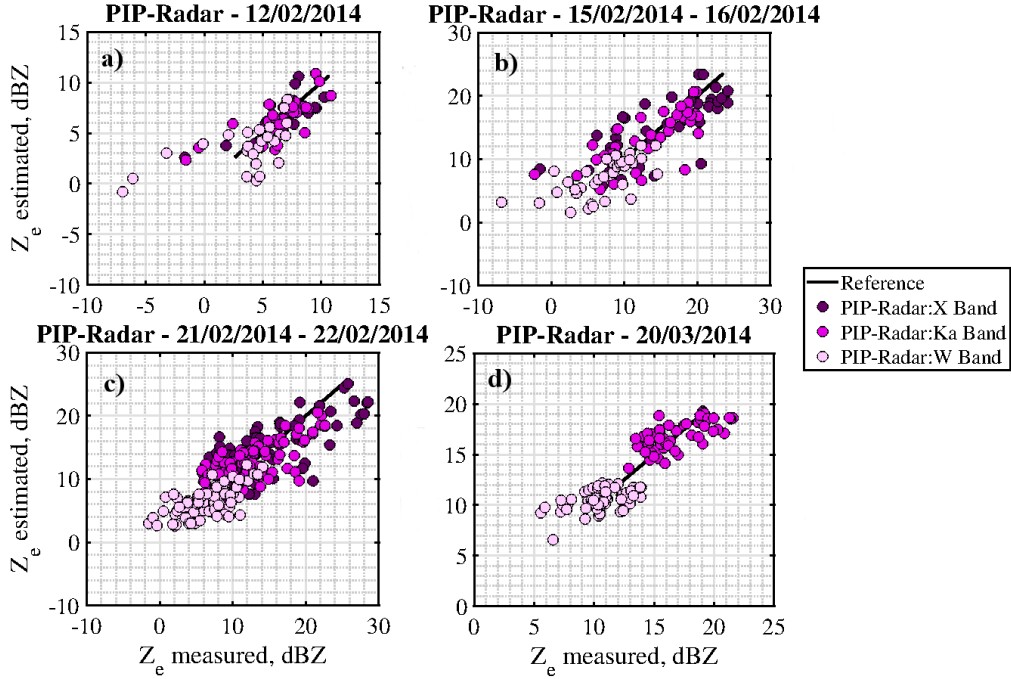

**Figure 7.** Comparison of radar reflectivity factor $Z_e$, measured by ARM radars at different frequencies (X-, Ka-, W-band), with $Z_e$ estimated from PIP using the assumed power-law form $Z_e = aS^b$. The liquid-water-equivalent snow rate $S$ in $\mathrm{mm/h}$ is evaluated from PIP, whereas the coefficients $a$ and $b$ are taken from Table 2.





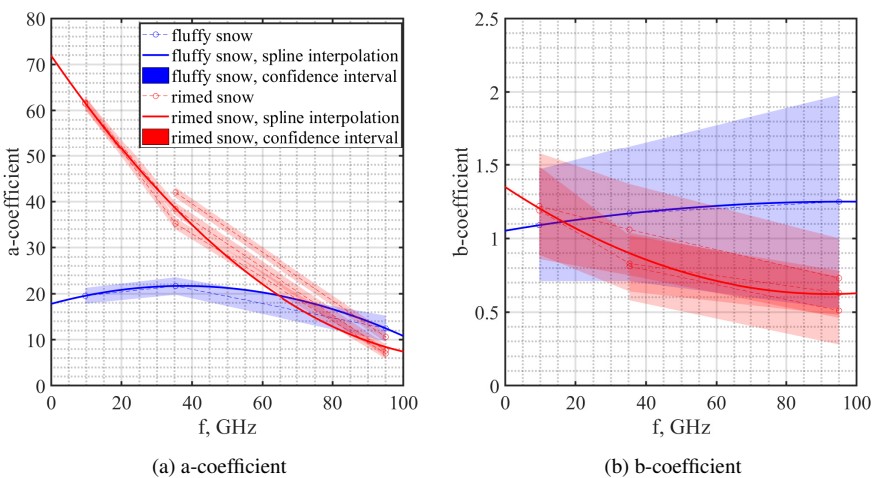

(a) a-coefficient          (b) b-coefficient

**Figure 8.** Frequency trend for the $a$ (left panel) and $b$ (right panel) regression coefficients, estimated in Table 2 using the power-law form $Z_e = aS^b$ and all BAECC collected data.





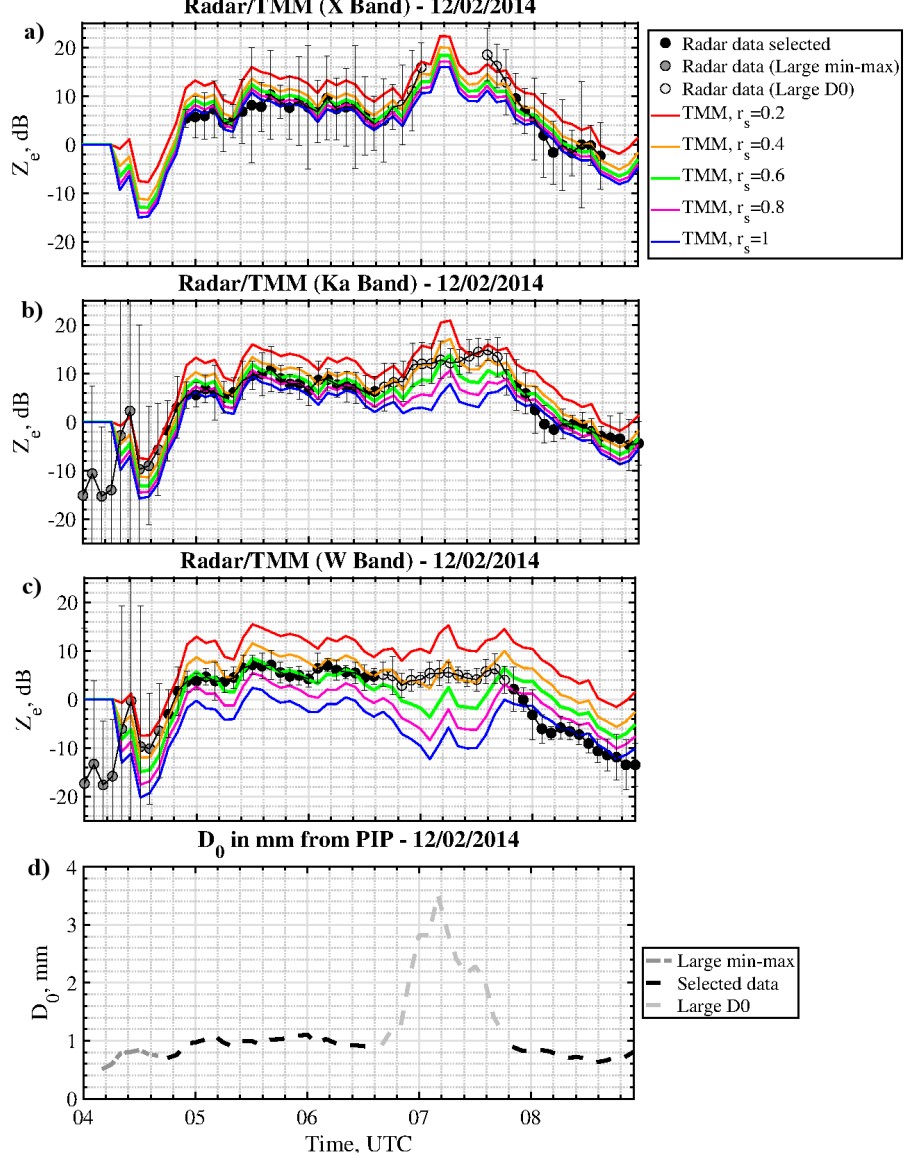

**Figure 9.** Radar and TMM computations from 12 February 2014 between 04:00 and 08:50 UTC (Fluffy snowfall). Radar reflectivities (black points) from XSACR, KAZR and MWACR are corrected for sky-noise, calibration offsets and attenuations (as better explained in Section 2.3). The error bars are used to represent the variation (min-max difference) of radar data within a 5-minute window with respect to their averaged value (black points). TMM-based computations (red, orange, green, magenta, blue lines for $r_s = 0.2$, $r_s = 0.4$, $r_s = 0.6$, $r_s = 0.8$ and $r_s = 1$, respectively) are derived from PIP data. The last panel shows the time series of the median volume diameter $D_0$ obtained from PIP. The time series is divided in three intervals: large $D_0$ (light grey points), large variation of the error bar (grey points) and the selected remaining radar data (black points).





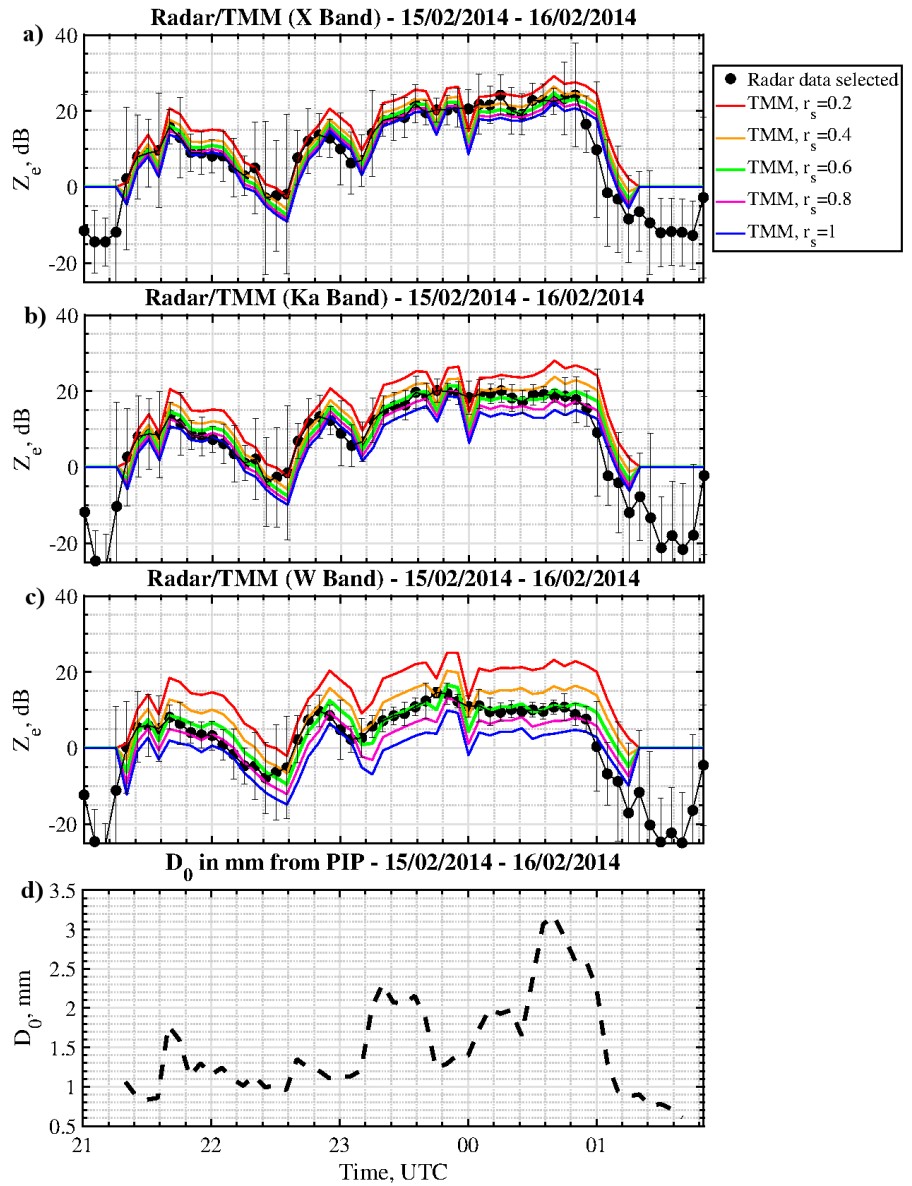

**Figure 10.** Same as Figure 9, but for 15/16 February 2014, 21:00-01:48 UTC (Rimed snowfall).





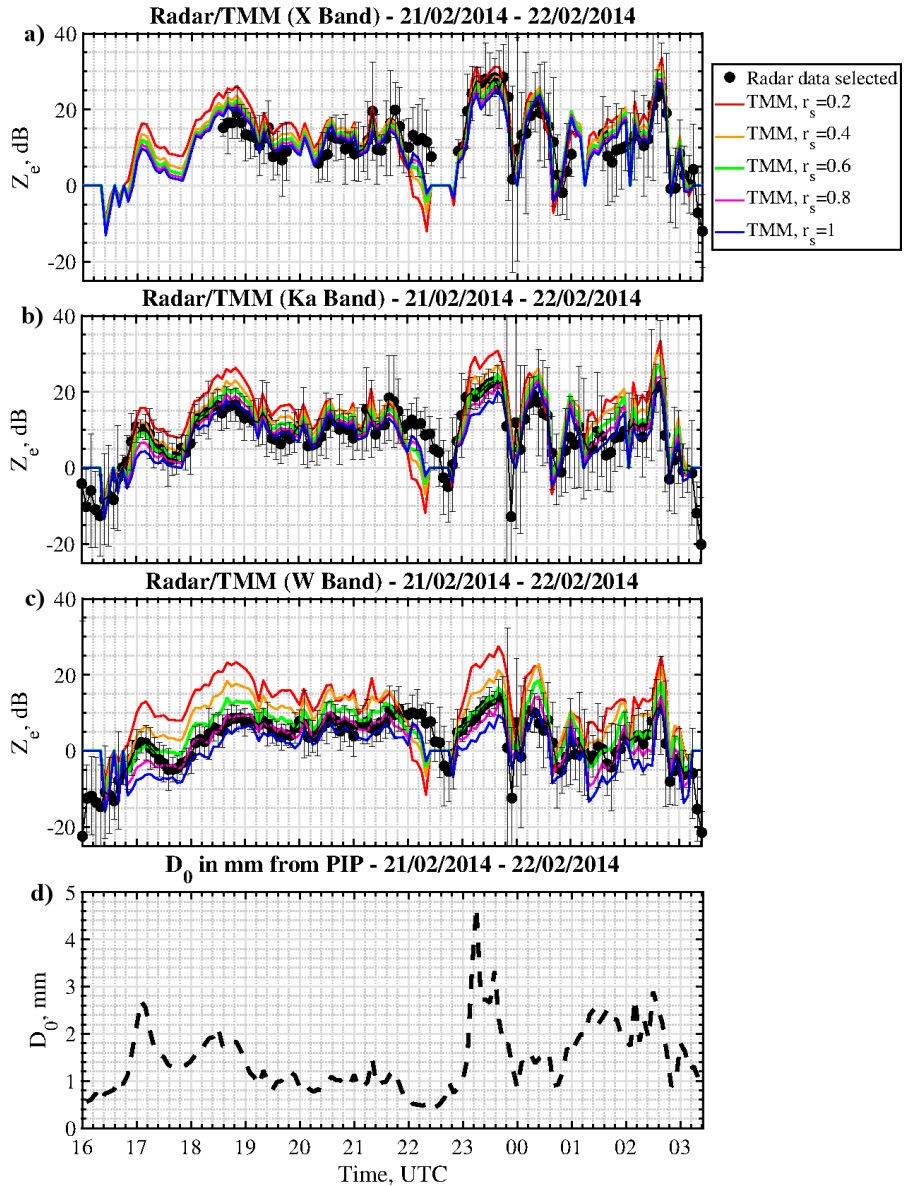

**Figure 11.** Same as Figure 9, but for 21/22 February 2014, 16:00-03:24 UTC (Rimed snowfall).





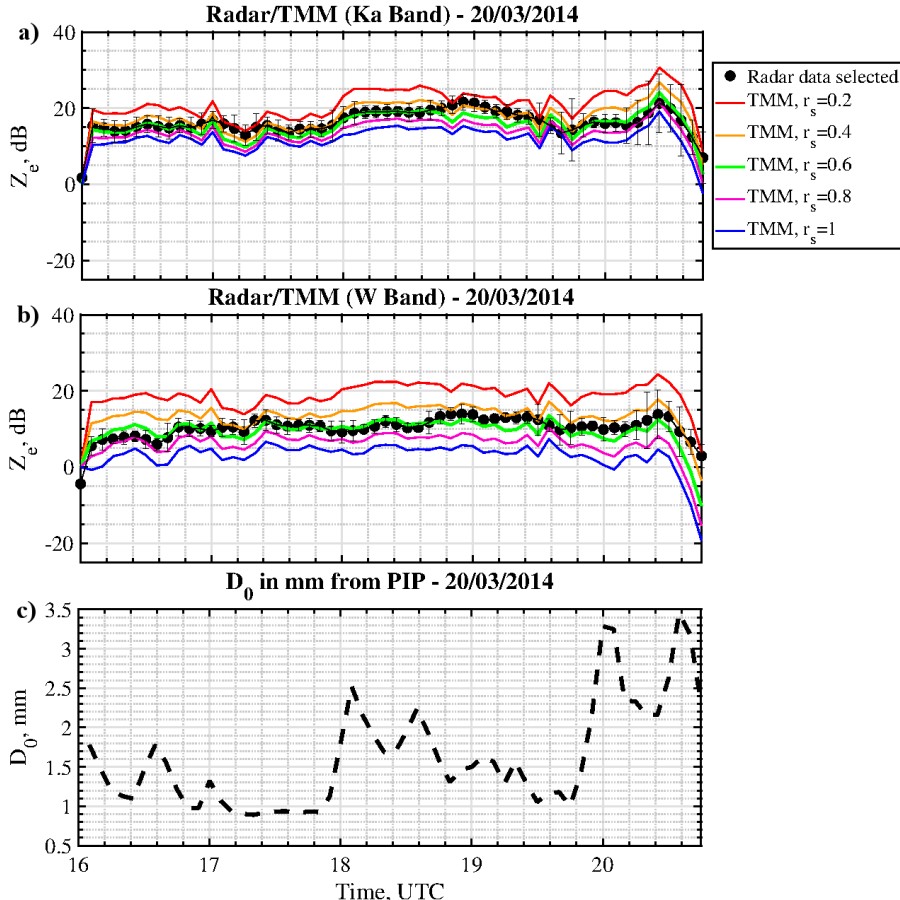

**Figure 12.** Same as Figure 9, but for 20 March 2014, 16:00- 20:48 UTC (Rimed snowfall). The X band radar data are not available for this time-window.



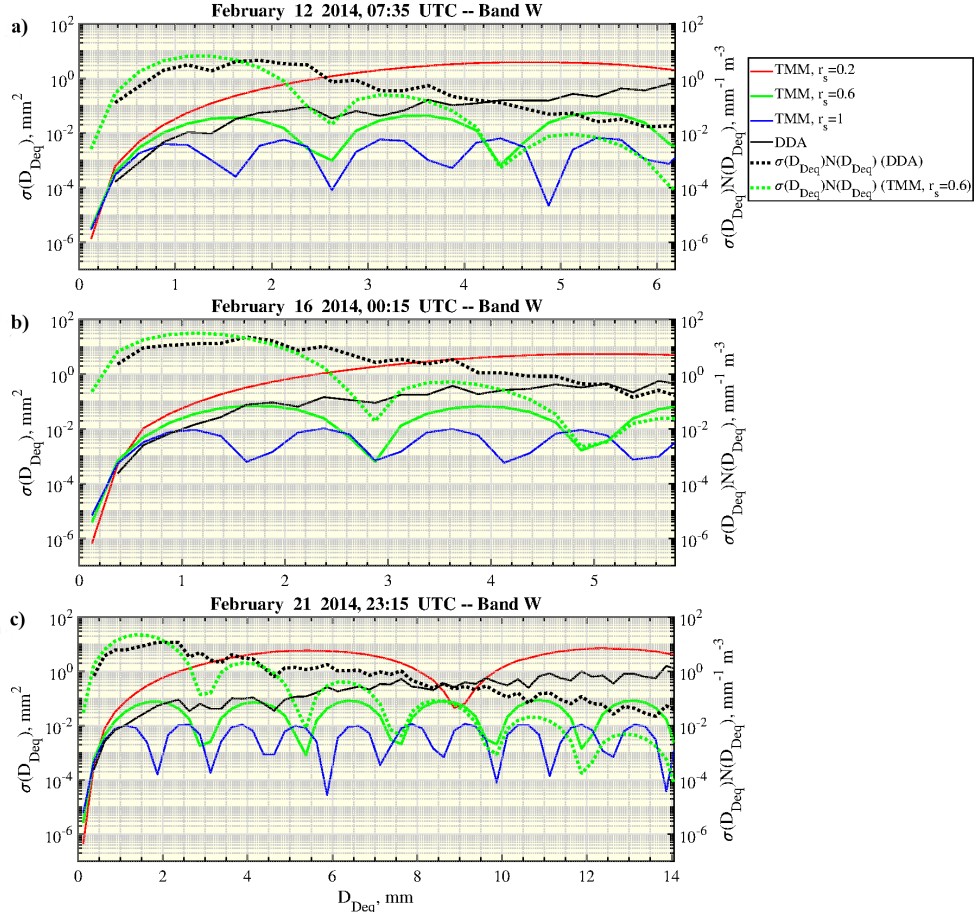

**Figure 13.** Horizontally-polarized cross section $\sigma$, expressed as a function of the diameter disk-equivalent $D_{Deq}$ at W band by comparing DDA computations (black line) and TMM computations (red, green and blue lines, matching $r_s = 0.2$, $r_s = 0.6$ and $r_s = 1$). The product between $\sigma$ and PIP-derived snowflake size distribution $N$ shows the main contribution of particle size in terms of diameter disk-equivalent for DDA computations (dotted black line) and TMM computations ($r_s = 0.6$, dotted green line).