# Peer review of "Snowfall retrieval at X, Ka and W bands: consistency of backscattering and microphysical properties using BAECC ground-based measurements"

_Atmospheric Measurement Techniques, 2017_

## Referee Comment (RC1) · Anonymous Referee #1 · 25 Jan 2018

The manuscript presents observed power-law relationships between liquid-water-equivalent snowfall rate S and radar equivalent reflectivity factor Ze at X, Ka, and W frequency bands for four snowfall events. They found that the power-law Ze-S relationships are distinguishable between fluffy and rimed snowfall events. To better understand the connection of snowflake microphysics with their scattering properties, numerical scattering calculations were conducted using both soft-spheroid (TMM) and detailed (DDA) ice particles, with mass and size constrained by PIP measurements. They argued that soft-spheroid approximation overestimates the back-scattering cross

sections of small ice particles, but underestimates those of large ice particles; on average, soft-spheroid approximation with proper aspect ratio explains the observed Ze-S relationships.

Major comments:

I believe this paper represents a substantial contribution in not only collocating multiple-frequency radar observations with in-situ image measurements of ice particles, but also exploring the capacity of numerical scattering simulations with simplified spheroid models. However, a more detailed analysis of the four cases to show the clear difference between the two precipitation modes is needed. Furthermore, a discussion on the physical reasons of separating into such two precipitation modes would be more valuable. The specific major comments are as follows.

1. It is not clear what the definition of fluffy snowfall and rimed snowfall is. Based on the paper, fluffy snowflakes refer to small low-density ice particles, while rimed snowflakes refer to large high-density ice particles. However, low-density ice particles can be large if there is a high number concentration of ice crystals and they aggregates to large particles. Riming occurs when ice particles collect super-cooled cloud drops through a super-cooled liquid layer. So density can probably separate fluffy and rimed snowflakes, but not size. Please provide more information and evidence, e.g., PIP images, about the details on what exactly separate the two precipitation modes.

2. Discuss why the two precipitation modes have such a difference in a and b coefficients in the Ze-S relationship?

3. Page 13 line 12: "The latter consideration leads to the conclusion that the soft-spheroid approximation may work rather well for computing radar reflectivity since the errors for larger particles are compensated by those for smaller particles". This conclusion is very questionable, because particle size distribution (PSD) does change and it changes the weight between small and large particles. The error might cancels out in specific cases, but not always.

**AMTD**

4. Can you add the results from DDA simulation in Figs. 3-6 and 9-12? DDA simulation is only discussed at the end in Fig. 13 in terms of backscatter cross section as a function of size. It will be great to see how the detailed ice particles match with observations.

Minor comments:

1. Page 7 line 22-23: 'This is because the microwave backscatter properties do not depend on the small details, but mostly on the overall structure, at least at cm-wavelength'. This is not true. Backscatter cross section does depend on the details of the structure even at large wavelength.

2. Page 7 line 26: typo "looses" » "loses".

3. Page 7 line 27: typo "dendrities" » "dendrites".

4. Page 7 line 29-33: This sentence is not clear. Please revise.

5. Page 8 line 10: Dmax is obtained from PIP. In page 4 line 1, the disk-equivalent diameter DDeq is also obtained from PIP. Are they related? And how?

6. Page 9 last paragraph: The particles are randomly oriented from DDA calculations, while the spheroids of TMM are oriented horizontally with 10° standard deviation from Page 8 line 3. Please comment on how the inconsistency affects the scattering results.

7. Page 11 line 20: typo. "cleare" » "clear".

8. Page 12 line 19: typo. Remove "the" in "For the this case ...".

9. Page 12 line 23: typo. Remove "is" or "equals to" in "... is on an average equals to ...".

---

## Referee Comment (RC2) · Anonymous Referee #2 · 30 Jan 2018

The author developed the observed relationships between snow rate (S) and radar reflectivity factor (Ze) by combing in situ measurements and radar measurements at X, Ka and W bands. From the selected four snow cases, it was found that the Ze-S relationships for fluffy snowflakes are different from those for rimed snowflakes. The scattering simulations were also conducted using the TMM and DDA methods. The author concluded that the TMM method is suitable for radar reflectivity simulations by choosing the optimal aspect ratio which is shown in this paper for different frequencies and snowflake habits. The most contribution from this paper is to find the optimal

aspect ratio for fluffy and rimed snowflakes at X, Ka and W bands, which can be used in developing the snowfall retrieval algorithms using radar measurements. However, some methods in processing the data and discussions need improvements or revisions.

Major Comments: 1. The author used the fixed calibration offsets for the snowfall experiments, which is not reasonable. Since those observed Ze-S relationships are the reference relationships for selecting the optimal aspect ratios, it is important to correct the errors in radar reflectivity considerably. The attenuation at Ka and W bands due to the liquid water and snow can be significant and is heavily profile-dependent. We need to calculate the attenuation at Ka and W bands due to the cloud liquid water and snow for each profile, even the author only used the near-surface bin. I understand that the reliable source of cloud liquid water profile might not be available for the datasets used in this paper, but we should at least correct the attenuation due to the snow using a better method. See the reference: Kulie, M. S., M. J. Hiley, R. Bennartz, S. Kneifel, and S. Tanelli (2014), Triple frequency radar reflectivity signatures of snow: Observations and comparisons to theoretical ice particle scattering models, J. Appl. Meteorol. Climatol., 1080–1098, doi:10.1175/JAMC-D-13-066.1.

2. Please clarity the definitions of "fluffy" and "rimed" snowflakes and why the author separated the snow events into these two types? Did the author try to study the "unrimed" and "rimed" snowflakes"? the "rimed" snowflakes are usually associated with high density, while the "unrimed" snowflakes can be considered as low-density particles. In this way, it is better to explain why two snowflake habits have different Ze-S relations.

3. Do you have the Ze-S relationships for DDA results? Since you choose the riming particle model, it is good to compare the DDA results using the riming particle model with the TMM results and the observations. Please add the DDA results for Fig. 3 to 6.

4. "The latter consideration leads to the conclusion that the soft spheroid approximation may work rather well for computing radar reflectivity since the errors for larger particles

are compensated by those for smaller particles". This conclusion is not correct, if you restrict the particle size range, you usually don't see this compensation.

Minor Comments: 1. Page 2, line 35, "from from", delete one 2. Page 3, line 31, change "64x48" to "64×48"

―――――――――――――――

---

## Referee Comment (RC3) · Anonymous Referee #3 · 30 Jan 2018

This manuscript presents Ze-S relations derived based on observational data of radar reflectivities at three frequencies and concurrent measurements of snowflake size distributions and snowfall accumulation rates. The observationally-based Ze-S relations are compared to the modeled ones using different scattering models and snowflake shape assumption. The paper contains useful practical information about multi-frequency Ze-S relations and also provides interesting results on comparing TMM and DDA based approaches for deriving backscatter properties of snowflakes. I would recommend the manuscript for publication after revision. During the revision process,

please address the comments given below.

General comments

1. You consider several rimed snowfall cases, but only one fluffy snowfall. I think that based on only one case, it is premature to make a conclusion that the coefficients in the fluffy snowfall Z-S relations have different from rimed snowfall frequency tendencies (Page 11, lines 29-31).

2. Radar calibration issues. Section 2.3. How did you ensure resolution volume collocation from the vertically pointing radars and the scanning C-band radar at cloud top where Rayleigh scattering is assumed for all frequencies? What about the absorption in supercooled liquid which is different at different frequencies?

3. Why did you use the gamma size distribution model (Page 8) rather than directly using PIP observed size distributions expressed in snowflake size bins?

4. It would be helpful if, for each frequency, the authors provide figures showing your best Ze-S relations (given in bold font in Table 3 for individual snowfall events) and some previous relations from literature. You cite a number of such relations for W and Ka-bands. For X-band also there have been a fair amount of previous studies (for example, Boucher and Wieler Journal of Climate and Applied Meteorology 1985, p.68; Fujiyoshi et al. JAM 1990, p. 147; Matrosov et al. JTECH 2009, p.2324; Huang et al., C-band, JTECH 2010, p. 637).

5. It would be interesting to know if Ze-S relations derived for the IKA C-band frequency would be much different from those at X-band?

Specific comments

1. Page 5 line 16: It is stated that ARM radar measurements were corrected for attenuation. Is it attenuation due to accumulated snow on the radome or attenuation in falling snow?

2. It is not clear if in your modeling you assumed the preferential orientation of the particles (Page 8, lines 1-5) or random orientation (Page 9, lines27-31). I do not understand your term "randomly orientated particles at fixed orientation". Please clarify.

3. Fig. 8: What coefficients are shown in Fig. 8? Are those corresponding to the dashed black lines in Figs. 3-7? Or something else?

4. Can you provide in Table 2 coefficients corresponding to the dashed black lines in Figs. 3-7?

5. How did you obtain Dmax from the disk equivalent PIP measurements of Ddeq ?

6. Fig. 9. What is D0 in this figure? Is it the same as given by eq. (6)?

7. Page 4 line 23: mm of water?

8. Radar calibration: As the IKA radar has a vertical resolution of about 1 km at the ARM site (∼1 deg @ 64 km) did you averaged vertically ARM radar measurements in vertical to match this resolution?

---

## Referee Comment (RC4) · Anonymous Referee #4 · 31 Jan 2018

This manuscript presents a very thorough comparison of snowfall measurements conducted at X, Ka and W radar frequency, with the interesting idea of identifying an optimal aspect ratio for each of the frequencies under investigation. The research topic is important, mostly but not only because of the upcoming launch of EarthCare. The manuscript is well written and easy to read. I therefore recommend publication after a few minor corrections.

**Major comments**

My only major comment is about the classification of snow, as either fluffy or rimed. Further details should be given about how this distinction is made, and propose for example some shape descriptors to discriminate the transition. This appears as the only major subjective choice to be motivated. I suggest a piece of literature on the subject: "Solid hydrometeor classification and riming degree estimation from pictures collected with a Multi-Angle Snowflake Camera", by Praz et al, AMT 2017. In this study, the authors presented a classification method that tried to be as much as possible in line with the classification of Magono and Lee (1966).

- Page 4: Could you add a sentence summarizing the possible limitations/error sources of PIP? (i.e. beef up the final sentence about the wind)
- Page4, Line 14: add the percentage of "rejected" particles for this specific campaign, if applicable
- Page 5, Line 24: add an error measure (standard deviation) of such intercomparison
- Page 7, line 5: could you elaborate also in term of sampling volume sizes, other than time?
- Page 8, Line 15: as a curiosity, did you perform any evaluation about the goodness of fit?

Typos Page 2, I.35: typo "from from"

---

## Referee Comment (RC5) · Anonymous Referee #5 · 7 Feb 2018

In this manuscript, the authors use collocated measurements of triple frequency vertically pointing radar measurements of snowfall with surface PIP PSD measurements. Using these collocations, they evaluate T-matrix method (TMM) simulations of snowfall for different snowfall types (fluffy, rimed) to determine the parameterizations that lead to the closest matches to measured reflectivities at different wavelengths.

There are few studies available that directly compare the differences in reflectivity-snowfall relationships at three frequencies, and fewer still that do so with measured data. This paper could be a valuable contribution towards the effort to find simple

calculations for the complex relationships between snowflake PSDs and reflectivity, but the result are based on an ambiguous definition of aspect ratio that appears both subjective to the radar and objective as a particle property (explained in the comments). With some clarification on the language, I would support the publication of this article.

Major comments:

I'm having trouble understanding how I'm supposed to view a particle's aspect ratio (rs ). On one hand, rs appears to be a real, measurable property of a particle. It is defined by a major and minor axis (page 7, line 24), and different aspect ratios refer to different specified particle geometries (page 7 lines 25 and 28; page 12 line 6). Throughout the paper, however, rs is also defined and used as a variable tuning parameter that can change for a given PSD depending on the radar frequency (page 12 lines 8-10). If rs signifies a particle shape, than the assumption of that shape shouldn't be able to change depending on the radar being used to observe it. If rs is intended as a tuning parameter, the language in the paper should be clear prevent any interpretations that the rs recommended could represent physical particle properties.

In Section 2.3, the authors claim "The cross-calibration method is based on the assumption that in the low reflectivity region at the cloud top the small crystals basically scatter in the Rayleigh regime (Hogan et al., 2000). In these regions, therefore, the measured radar reflectivity values from by all millimeter wave radars should match". These values may not match if there is substantial liquid water present, and liquid water is common in snowing clouds (Wang et.al 2014). Liquid water attenuation is very difficult to predict at different frequencies for supercooled liquid water droplets (Kneifel et. al 2014), and liquid water is also very hard to measure, so this attenuation may not be possible to fully address. But it should be discussed and, if possible, estimated.

Specific Comments:

Numerous spelling and grammar errors throughout. Suggest closer proofreading before final submission.

Is it necessary to include the information on the Pluvio gauge in 2.1? I don't see the data used in any of the figures.

Kneifel, S., Redl, S., Orlandi, E., Löhnert, U., Cadeddu, M. P., Turner, D. D., & Chen, M. T. (2014). Absorption properties of supercooled liquid water between 31 and 225 GHz: Evaluation of absorption models using ground-based observations. Journal of Applied Meteorology and Climatology, 53(4), 1028–1045. http://doi.org/10.1175/JAMC-D-13-0214.1

Wang, Y., Liu, G., Seo, E. K., & Fu, Y. (2013). Liquid water in snowing clouds: Implications for satellite remote sensing of snowfall. Atmospheric Research, 131, 60–72. http://doi.org/10.1016/j.atmosres.2012.06.008
* * *

---

## Short Comment (SC1) · 8 Feb 2018

The paper and measurements presented are interesting but there is one important conclusion that is not adequately supported by quantitative analysis.

In particular, when discussing figure 13, the authors conclude:

"... i) looking at the product between the PIP-derived PSD and the radar cross section $\sigma$, we note that the TMM-based product is higher than the DDA one for small ice particles and is lower for the larger particles. The latter consideration leads to the conclusion that

the soft-spheroid approximation may work rather well for computing radar reflectivity since the errors for larger particles are compensated by those for smaller particles."

This could not be true in general. It depends by the extreme of integration in terms of particle's diameter of the quantity $\sigma$*N(D) shown in figure 13 on the right side axis. If you integrate between 0 and 2.5 mm you will probably have a sort of compensation effect. This not likely happens if you consider larger integration limits. Unfortunately, the Authors do not report a figure where they show a statistic of N(D) measured from PIP to have an idea of typical show particle's range diameters for the considered case studies. They should add N(D) figure.

Minor: - why in figure 13, bottom panel Deq extends up to 14 mm whereas in the other panel it is up to 6 mm? - why DDA simulations starts from Deq =0.4 mm whereas TMM starts from approximatively 0.05 mm? Differences of $\sigma$*N(D) in that range of diameter can play a role.

---

## Author Comment (AC1) · 19 Apr 2018

We thank the reviewer for his suggestions and, in particular, for specific prompts to clarify some fundamental issues. Our detailed replies can be found below in after the "REPLY." label. Changes in the manuscript are highlighted in blue text.

Major comment: The paper and measurements presented are interesting but there is one important conclusion that is not adequately supported by quantitative analysis. In particular, when discussing figure 13, the authors conclude: "... i) looking at the product

between the PIP-derived PSD and the radar cross section $\sigma$, we note that the TMM-based product is higher than the DDA one for small ice particles and is lower for the larger particles. The latter consideration leads to the conclusion that the soft-spheroid approximation may work rather well for computing radar reflectivity since the errors for larger particles are compensated by those for smaller particles." This could not be true in general. It depends by the extreme of integration in terms of particle's diameter of the quantity $\sigma$*N(D) shown in figure 13 on the right side axis. If you integrate between 0 and 2.5 mm you will probably have a sort of compensation effect. This not likely happens if you consider larger integration limits. Unfortunately, the Authors do not report a figure where they show a statistic of N(D) measured from PIP to have an idea of typical show particle's range diameters for the considered case studies. They should add N(D) figure.

REPLY. Thank you for the consideration. We have forced the conclusion, our final assessment is valid but only related to our dataset. Now we have changed the conclusion also looking at the new modified manuscript in which we are using microwave observations of liquid water path ( LWP) to separate events into lightly, moderately rimed and heavily rimed snow. Thank you also to highlight the need to add a figure on N(D). We have integrated the TMM between 0 and 2.5D0 mm and this was not justified indeed. In the revised paper we have added Figure 13 in which we show the difference between the measured N(D), the estimated Gamma N(D) and the estimated Gamma N(D) truncated at 2.5 multiplied for D0. From Figure 13, respectively for (a) 12 February and (b) 15/16 February, it is noted that there is an under-estimation of the PSD for higher diameter.

Minor comments: 1 why in figure 13, bottom panel Deq extends up to 14 mm whereas in the other panel it is up to 6 mm?

REPLY. Figure 13 (now Figure 12) shows horizontally-polarized cross-section modelled by TMM and DDA but the diameter disk-equivalent used to set up the numerical simulations is from PIP data, then the maximum value of 2.5*DDeq depends from the

observed particles. Now we have changed the maximum value at DDeq=6 mm to fixed the scale limit.

2 why DDA simulations starts from Deq =0.4 mm whereas TMM starts from approximatively 0.05 mm? Differences of $\sigma$*N(D) in that range of diameter can play a role.

REPLY. As explained in section 3.3 the DDA cross sections are computed by averaging particle properties within each bin of the PIP measured PSD and plotted at the bin center. Minimum bin center was 0.375 mm which is representative of particles with sizes ranging from 0.250 to 0.5 in size. However, thanks to the reviewer suggestion, we have extended the plot of DDA scattering cross section down to particles of 0.125 mm in size as they are plotted for the TMM quantities for an easier comparison.

Thank you again for the questions, the supplement to this comment contains the revised AMT manuscript. Changes in the manuscript are highlighted in blue text.

Please also note the supplement to this comment:
https://www.atmos-meas-tech-discuss.net/amt-2017-485/amt-2017-485-AC1-
supplement.pdf

---

## Author Comment (AC3) · 19 Apr 2018

We thank the reviewer for his suggestions and, in particular, for specific prompts to clarify some fundamental issues. Our detailed replies can be found below in after the "REPLY." label. Changes in the manuscript are highlighted in blue text.

Major comments: 1 The author used the fixed calibration offsets for the snowfall experiments, which is not reasonable. Since those observed Ze-S relationships are the reference relationships for selecting the optimal aspect ratios, it is important to correct

the errors in radar reflectivity considerably. The attenuation at Ka and W bands due to the liquid water and snow can be significant and is heavily profile-dependent. We need to calculate the attenuation at Ka and W bands due to the cloud liquid water and snow for each profile, even the author only used the near-surface bin. I understand that the reliable source of cloud liquid water profile might not be available for the datasets used in this paper, but we should at least correct the attenuation due to the snow using a better method. See the reference: Kulie, M. S., M. J. Hiley, R. Bennartz, S. Kneifel, and S. Tanelli (2014), Triple frequency radar reflectivity signatures of snow: Observations and comparisons to theoretical ice particle scattering models, J. Appl. Meteorol. Climatol., 1080–1098, doi:10.1175/JAMC-D-13-066.1.

REPLY According to Kulie et al. (2014) the W-band attenuation due to snow ranges between 0.2 and 1 dB km$^{-1}$. Since, we are taking measurements close to the ground, and the expected attenuation is between 0.08 and 0.4 dB. Therefore, the attenuation due to snow can be ignored. The attenuation due to supercooled liquid water is expected to be 1 to 4 dB km$^{-1}$. That means that at maximum we expect the liquid water attenuation of around 1 dB. Given the uncertainty in the attenuation correction, we have decided not to apply it.

A potentially significant source of attenuation, is the radome attenuation. Because of this, the radar cross calibration was performed before and after the events and cases where these estimated values were different were ignored. Furthermore, the radar noise power was analyzed to identify radome attenuation.

2 Please clarity the definitions of "fluffy" and "rimed" snowflakes and why the author separated the snow events into these two types? Did the author try to study the "unrimed" and "rimed" snowflakes"? the "rimed" snowflakes are usually associated with high density, while the "unrimed" snowflakes can be considered as low-density particles. In this way, it is better to explain why two snowflake habits have different Ze-S relations.

REPLY. Thank you for pointing out the problem. In the modified manuscript we are using microwave observations of liquid water path ( LWP) to separate events into lightly, moderately rimed and heavily rimed snow. Even though LWP is not a direct measure of degree of riming, LWP and riming are related as shown for example in (Moisseev et al., 2017). We also agree with the fact that the original definition of rimed and unrimed snowfall was vague and not properly explained but now we have heavily modified it.

3 Do you have the Ze-S relationships for DDA results? Since you choose the riming particle model, it is good to compare the DDA results using the riming particle model with the TMM results and the observations. Please add the DDA results for Fig. 3 to 6.

REPLY. The comparison of TMM backscattering cross sections with DDA has been performed for validation purposes. We are aware of the limitations of TMM and then we wanted to check our results. However this comparison is not the central point of the study and we think that adding further curves to the plots would make them very confusing. On the other hand a parallel study is under preparation that further explore the link between the microphysical and scattering properties of snow where this comparison can be better addressed.

4 "The latter consideration leads to the conclusion that the soft spheroid approximation may work rather well for computing radar reflectivity since the errors for larger particles are compensated by those for smaller particles". This conclusion is not correct, if you restrict the particle size range, you usually don't see this compensation.

REPLY. We are not using unrestricted sizes, the particle size range is restricted to 2.5 $D_0$. Minor comments:

5 Page 2, line 35, from from, delete one

REPLY. Done.

6 Page 3, line 31, change 64x48 to 64$\times$48

REPLY. Done.

Thank you again for the questions, the supplement to this comment contains the revised AMT manuscript. Changes in the manuscript are highlighted in blue text.

Please also note the supplement to this comment:
https://www.atmos-meas-tech-discuss.net/amt-2017-485/amt-2017-485-AC3-supplement.pdf

—————————————————————

[Figure]

**Supplement:**

[revised manuscript text omitted]

---

## Author Comment (AC7) · 19 Apr 2018

We thank the reviewer for his suggestions and, in particular, for specific prompts to clarify some fundamental issues. Our detailed replies can be found below in after the "REPLY." label. Changes in the manuscript are highlighted in blue text.

Major comments:

1 I'm having trouble understanding how I'm supposed to view a particle's aspect ratio ($r_s$). On one hand, $r_s$ appears to be a real, measurable property of a particle. It

is defined by a major and minor axis (page 7, line 24), and different aspect ratios refer to different specified particle geometries (page 7 lines 25 and 28; page 12 line 6). Throughout the paper, however, rs is also defined and used as a variable tuning parameter that can change for a given PSD depending on the radar frequency (page 12 lines 8-10). If rs signifies a particle shape, than the assumption of that shape shouldn't be able to change depending on the radar being used to observe it. If rs is intended as a tuning parameter, the language in the paper should be clear prevent any interpretations that the rs recommended could represent physical particle properties.

REPLY. The aspect ratio is the parameter of the soft-spheroid particle model. It may or may not coincide with the measurable particle property. We have modified the text to make this point clearer.

2 In Section 2.3, the authors claim "The cross-calibration method is based on the assumption that in the low reflectivity region at the cloud top the small crystals basically scatter in the Rayleigh regime (Hogan et al., 2000). In these regions, therefore, the measured radar reflectivity values from by all millimeter wave radars should match". These values may not match if there is substantial liquid water present, and liquid water is common in snowing clouds (Wang et.al 2014). Liquid water attenuation is very difficult to predict at different frequencies for supercooled liquid water droplets (Kneifel et. al 2014), and liquid water is also very hard to measure, so this attenuation may not be possible to fully address. But it should be discussed and, if possible, estimated. Kneifel, S., Redl, S., Orlandi, E., LoÌˆhnert, U., Cadeddu, M. P., Turner, D. D., & Chen, M. T. (2014). Absorption properties of supercooled liquid water between 31 and 225 GHz: Evaluation of absorption models using ground-based observations. Journal of Applied Meteorology and Climatology, 53(4), 1028–1045. http://doi.org/10.1175/JAMC-D-13-0214.1 Wang, Y., Liu, G., Seo, E. K., & Fu, Y. (2013). Liquid water in snowing clouds: Implications for satellite remote sensing of snowfall. Atmospheric Research, 131, 60–72. http://doi.org/10.1016/j.atmosres.2012.06.008

REPLY. For cross calibration only non precipitating clouds with no or little supercooled

liquid water were used. Since we have used data from the lowest usable range gates, the expected liquid water attenuation should be less than 1 dB. That is why not liquid water or snow attenuation correction is applied.

Specific comments:

1 Numerous spelling and grammar errors throughout. Suggest closer proofreading before final submission.

REPLY. Thanks for the suggestions, we have heavily revised the manuscript by introducing some modifications highlighted in blue text within the revised text. We hope that the overall text revision is helpfully for a clearer understanding of the content.

2 Is it necessary to include the information on the Pluvio gauge in 2.1? I don't see the data used in any of the figures.

REPLY. Pluvio gauge has been used to check the snow rate from PIP and to estimate the LWE. We have added it in the revised paper a sentence on page 4 line 23-25.

Thank you again for the questions, the supplement to this comment contains the revised AMT manuscript. Changes in the manuscript are highlighted in blue text.

Please also note the supplement to this comment:
https://www.atmos-meas-tech-discuss.net/amt-2017-485/amt-2017-485-AC7-supplement.pdf

---

## Author Response (AR2)

**Final response to all referee comments (RCs)**

**We thank the Editors and the reviewers for their suggestions and, in particular, for specific prompts to clarify some fundamental issues. Our detailed replies can be found below in bold red text after the "REPLY." label. Changes in the manuscript are highlighted in blue text.**

**RC1**: 'Comments on "Snowfall retrieval at X, Ka and W band: consistency of backscattering and microphysical properties using BAECC ground-based measurements'", Anonymous Referee #1, 25 Jan 2018

*The manuscript presents observed power-law relationships between liquid-water- equivalent snowfall rate S and radar equivalent reflectivity factor Ze at X, Ka, and W frequency bands for four snowfall events. They found that the power-law Ze-S relationships are distinguishable between fluffy and rimed snowfall events. To better understand the connection of snowflake microphysics with their scattering properties, numerical scattering calculations were conducted using both soft-spheroid (TMM) and detailed (DDA) ice particles, with mass and size constrained by PIP measurements. They argued that soft-spheroid approximation overestimates the back-scattering cross sections of small ice particles, but underestimates those of large ice particles; on average, soft-spheroid approximation with proper aspect ratio explains the observed Ze-S relationships.*

v  Major comments:

Ø  General comment:

I believe this paper represents a substantial contribution in not only collocating multiple-frequency radar observations with in-situ image measurements of ice particles, but also exploring the capacity of numerical scattering simulations with simplified spheroid models. However, a more detailed analysis of the four cases to show the clear difference between the two precipitation modes is needed. Furthermore, a discussion on the physical reasons of separating into such two precipitation modes would be more valuable.

Ø  Specific comments:

1  It is not clear what the definition of fluffy snowfall and rimed snowfall is. Based on the paper, fluffy snowflakes refer to small low-density ice particles, while rimed snowflakes refer to large high-density ice particles. However, low-density ice particles can be large if there is a high number concentration of ice crystals and they aggregate to large particles. Riming occurs when ice particles collect super-cooled cloud drops through a super-cooled liquid layer. So density can probably separate fluffy and rimed snowflakes, but not size. Please provide more information and evidence, e.g., PIP images, about the details on what exactly separate the two precipitation modes.

**REPLY Thank you for pointing out the problem. We agree on the fact that the original definition of rimed and unrimed snowfall was vague and not properly explained. In the modified manuscript we are using microwave observations of liquid water path ( LWP) to separate events into lightly, moderately rimed and heavily rimed snow. Even though LWP is not a direct measure of degree of riming, LWP and riming are related as shown for example in (Moisseev et al., 2017).**

2  Discuss why the two precipitation modes have such a difference in a and b coefficients in the Ze-S relationship?

**REPLY As shown by von Lerber et al (2017), the prefactor of the instantaneous Ze-S relation depends on particle physical properties (expressed in terms of prefactor of RCS(D) relation) and intercept parameter of PSD. The exponent of Ze-S relation depends on the exponent of RCS(D) and the shape parameter of PSD. In the Rayleigh regime, the dependence of radar cross section on D, RCS(D), is given by (m(D))^2. It should be noted, that for the Ze-S relations derived for an event or averaged over several snowfall storms, the above-stated dependence becomes less clear because of changes in m(D) and PSD.**

**For higher radar frequencies, RCS(D) relation is no longer given by (m(D))^2. For example, the exponent of RCS(D) will become smaller.  Also the prefactor would change. These changes explain changes in Ze-S, as we go from one frequency to another. However, the observed difference is also caused by changes in PSD, and RCS(D), during the events. The variability in PSD and RCS(D) is probably different for different snowfall type. At the moment, we cannot separate the effects and it is not clear what is the main cause for the changes in a and b coefficients. However, it appears that as we use higher radar frequency the difference between Ze-S prefactors for different snow types becomes smaller.**

3   Page 13 line 12: "The latter consideration leads to the conclusion that the soft-spheroid approximation may work rather well for computing radar reflectivity since the errors for larger particles are compensated by those for smaller particles". This conclusion is very questionable, because particle size distribution (PSD) does change and it changes the weight between small and large particles. The error might cancel out in specific cases, but not always.

**REPLY We agree with the reviewer's comment. But in the order to study the impact of the assumed scattering model on retrievals, studies similar to the presented one is needed.  For example, it could turn out that given almost exponential PSD and m(D)~D^2, the observed compensating effect is common. The current analysis is limited and we agree that more studies are needed.**

4   Can you add the results from DDA simulation in Figs. 3-6 and 9-12? DDA simulation is only discussed at the end in Fig. 13 in terms of backscatter cross section as a function of size. It will be great to see how the detailed ice particles match with observations.
**REPLY. The comparison of TMM backscattering cross sections with DDA has been performed for validation purposes. We are aware of the limitations of TMM and then we wanted to check our results. However this comparison is not the central point of the study and we think that adding further curves to the plots would make them very confusing. On the other hand a parallel study is under preparation that further explore the link between the microphysical and scattering properties of snow where this comparison can be better addressed.**

v  Minor comments:

1 Page 7 line 22-23: 'This is because the microwave backscatter properties do not depend on the small details, but mostly on the overall structure, at least at cm-wavelength'. This is not true. Backscatter cross section does depend on the details of the structure even at large wavelength.

**REPLY. Thank you for the comment. Indeed, we have wrongly used the verb "depend". The aim was to express that at centimetre and millimetre-wave radar frequencies the small details in a particle structure usually do not significantly affect the backscatter properties. The latter depend largely on the overall shape, which, in the case of spheroid, is determined by the spheroid aspect ratio, rs (Matrosov, 2007; Dungey and Bohren, 1993). In the revised paper we have removed the sentence and explained in more details way we have used the TMM (Page 8 line 1-13).**

2 Page 7 line 26: typo "looses" » "loses".

**REPLY. Done.**

3 Page 7 line 27: typo "dendrities" » "dendrites".

**REPLY. Done.**

4 Page 7 line 29-33: This sentence is not clear. Please revise.

**REPLY. The soft-spheroid, used in TMM, and complex particles, used in DDA computations, are particle models. Those are not real particles, but our representations of those. As in all models, there are tuning parameters that need to be adjusted to match the observations. We should note, that reproducing the physical appearance of snowflakes is not one of the goals (at least, not the most important goal) of using such models in microwave remote sensing applications. We need a model that links precipitation rate, IWC, Dm, etc. and radar observations. The soft-spheroid model used in this study, is based on observations of m(D). The observed m(D) is our link to precip. rate. The particle aspect ratio and orientation are free parameters. The particle aspect ratio is a particularly important parameter, because it controls density and therefore the refractive index. More discussion on the topic is added to the text (in particular on page 8 line 14-17).**

5 Page 8 line 10: Dmax is obtained from PIP. In page 4 line 1, the disk-equivalent diameter DDeq is also obtained from PIP. Are they related? And how?

**REPLY: Because of the pre-defined parameter selection with the PIP instrument, the disk-equivalent diameter is recorded. However, in von Lerber et al. 2017, the maximum diameter for each particle is defined by fitting an ellipse to the measured bounding box considering also the orientation of the particle in respect to horizontal direction. The maximum value Dmax of the several observed maximum diameter values is saved. A linear conversion factor between Dmax and DDeq is defined for each snowfall event, and as stated in von Lerber et al. 2017, the value is deviating between 1.20-1.51 and the mean value is 1.38. In the revised paper we added clarification on page 4 line 13-15.**

6    Page 9 last paragraph: The particles are randomly oriented from DDA calculations, while the spheroids of TMM are oriented horizontally with 10° standard deviation from Page 8 line 3. Please comments on how the inconsistency affects the scattering results.

**REPLY In this study scattering database for rimed snowflakes by Leinonen and Szyrmer (2015) is used. They have achieved preferential alignment of snowflakes as follows:** *"To simulate the partial horizontal alignment of snowflakes in the atmosphere, the shortest principal axis of each aggregate is aligned at a normally distributed random angle, with a mean of 0 and a standard deviation of 40"*. **Therefore, both soft-spheroid and complex particles are preferentially aligned horizontally. However, their orientation angle distributions and, probably, aspect ratios do not necessary match. It is possible that the soft-spheroid model needed to fit radar observations does not represent exactly geometrical properties of snowflakes. It is also possible that the complex snowflake model is not physically correct. From the radar remote sensing perspective, if both models are consistent with the radar observations then both particle models are correct.**

**In this study we are introducing one of the methods to judge applicability of different scattering models. Of course, the present dataset is limited and more studies in this direction is needed. We have also added more explanation on Section 3.2 and 3.3.**

7 Page 11 line 20: typo. "cleare" » "clear".
**REPLY. Done.**
8  Page 12 line 19: typo. Remove "the" in "For the this case ...".
**REPLY. Done.**
9  Page 12 line 23: typo. Remove "is" or "equals to" in "... is on an average equals to ...".
**REPLY. Done.**

**RC2:** 'Comments on "Snowfall retrieval at X, Ka and W band: consistency of backscattering and microphysical properties using BAECC ground-based measurements"', Anonymous Referee #2, 30 Jan 2018

*The author developed the observed relationships between snow rate (S) and radar reflectivity factor (Ze) by combing in situ measurements and radar measurements at X, Ka and W bands. From the selected four snow cases, it was found that the Ze-S relationships for fluffy snowflakes are different from those for rimed snowflakes. The scattering simulations were also conducted using the TMM and DDA methods. The author concluded that the TMM method is suitable for radar reflectivity simulations by choosing the optimal aspect ratio which is shown in this paper for different frequencies and snowflake habits. The most contribution from this paper is to find the optimal aspect ratio for fluffy and rimed snowflakes at X, Ka and W bands, which can be used in developing the snowfall retrieval algorithms using radar measurements. However, some methods in processing the data and discussions need improvements or revisions.*

v Major comments:

1 The author used the fixed calibration offsets for the snowfall experiments, which is not reasonable. Since those observed Ze-S relationships are the reference relationships for selecting the optimal aspect ratios, it is important to correct the errors in radar reflectivity considerably. The attenuation at Ka and W bands due to the liquid water and snow can be significant and is heavily profile-dependent. We need to calculate the attenuation at Ka and W bands due to the cloud liquid water and snow for each profile, even the author only used the near-surface bin. I understand that the reliable source of cloud liquid water profile might not be available for the datasets used in this paper, but we should at least correct the attenuation due to the snow using a better method. See the reference: Kulie, M. S., M. J. Hiley, R. Bennartz, S. Kneifel, and S. Tanelli (2014), Triple frequency radar reflectivity signatures of snow: Observations and comparisons to theoretical ice particle scattering models, J. Appl. Meteorol. Climatol., 1080–1098, doi:10.1175/JAMC-D-13-066.1.

**REPLY According to Kulie et al. (2014) the W-band attenuation due to snow ranges between 0.2 and 1 dB km^-1. Since, we are taking measurements close to the ground, and the expected attenuation is between 0.08 and 0.4 dB. Therefore, the attenuation due to snow can be ignored. The attenuation due to supercooled liquid water is expected to be 1 to 4 dB km^-1. That means that at maximum we expect the liquid water attenuation of around 1 dB. Given the uncertainty in the attenuation correction, we have decided not to apply it.**

**A potentially significant source of attenuation, is the radome attenuation. Because of this, the radar cross calibration was performed before and after the events and cases where these estimated values were different were ignored. Furthermore, the radar noise power was analyzed to identify radome attenuation.**

We did perform the radar cross calibration for all the events. Because the values were similar, we have decided to apply only one set of values and not to change them.

In Figure 1 of the paper we represent the radar profiles for the 15 February 2014 at 17:13 UTC highlighting that:

$$\Delta\_(X-C)=2.94 \text{ dBZ}$$
$$\Delta\_(Ka-X)=1.33 \text{ dBZ}$$
$$\Delta\_(W-Ka)=0.89 \text{ dBZ}$$

and obtaining:

$$\Delta\_(Ka-C)=4.27 \text{ dBZ}$$
$$\Delta\_(W-C)=5.16 \text{ dBZ}$$

Now, to cross-validate these results, we can also show in Figure A the radar profiles for the 12 February 2014 at 04:28 UTC in which we confirm the previous miscalibrations values:

$$\Delta\_(X-C)=2.97 \text{ dBZ}$$
$$\Delta\_(Ka-X)=1.33 \text{ dBZ}$$
$$\Delta\_(W-Ka)=0.23 \text{ dBZ}$$

[Figure]

(a)

[Figure]

(b)

Figure A:

**We do not insert Figure A to avoid repetition but we have added more details about the cross-calibration in Section 2.4.**

2   Please clarity the definitions of "fluffy" and "rimed" snowflakes and why the author separated the snow events into these two types? Did the author try to study the "un-rimed" and "rimed" snowflakes"? the "rimed" snowflakes are usually associated with high density, while the "unrimed" snowflakes can be considered as low-density particles. In this way, it is better to explain why two snowflake habits have different Ze-S relations.

**REPLY. See reply above to the Comment n. 1-2 of the Reviewer n. 1.**

3   Do you have the Ze-S relationships for DDA results? Since you choose the riming particle model, it is good to compare the DDA results using the riming particle model with the TMM results and the observations. Please add the DDA results for Fig. 3 to 6.

**REPLY. See reply above to comment 4 reviewer n. 1**

4   "The latter consideration leads to the conclusion that the soft spheroid approximation may work rather well for computing radar reflectivity since the errors for larger particles are compensated by those for smaller particles". This conclusion is not correct, if you restrict the particle size range, you usually don't see this compensation.

**REPLY.  We are not using unrestricted sizes, the particle size range is restricted to 2.5 D0.**

Minor comments:

5   Page 2, line 35, "from from", delete one

**REPLY. Done.**

6   Page 3, line 31, change "64x48" to "64×48"

**REPLY. Done.**

**RC3:**'Review of the manuscript by Falconi et al. "Snowfall retrievals at X, Ka and W-band: consistency of backscattering and microphysical properties using BAECC ground-based measurements" amt-2017-485.', Anonymous Referee #3, 30 Jan 2018

*This manuscript presents Ze-S relations derived based on observational data of radar reflectivities at three frequencies and concurrent measurements of snowflake size distributions and snowfall accumulation rates. The observationally-based Ze-S relations are compared to the modeled ones using different scattering models and snowflake shape assumption. The paper contains useful practical information about multi-frequency Ze-S relations and also provides interesting results on comparing TMM and DDA based approaches for deriving backscatter properties of snowflakes. I would recommend the manuscript for publication after revision. During the revision process, please address the comments given below.*

v   General comments:

1   You consider several rimed snowfall cases, but only one fluffy snowfall. I think that based on only one case, it is premature to make a conclusion that the coefficients in the fluffy snowfall Z-S relations have different from rimed snowfall frequency tendencies (Page 11, lines 29-31).

**REPLY In the revised manuscript we split the cases according to LWP. We agree that more data is still needed to make a more solid statement, but we believe it can be quite representative as a preliminary attempt.**

2   Radar calibration issues. Section 2.3. How did you ensure resolution volume collocation from the vertically pointing radars and the scanning C-band radar at cloud top where Rayleigh scattering is assumed for all frequencies? What about the absorption in supercooled liquid which is different at different frequencies?

**REPLY The C-band radar was performing RHI scans over the site every 15 min. The observation from these observations are used for cross calibration. We have followed the approach of Hogan et al., 2000 selecting the region to statistical analysed the calibration manually by looking to the radar profiles at C-, X-, Ka- and W-band. We have compared the radar measurements in regions close to cloud top (height higher than 5 km) where we can expect ice hydrometeors to be mostly Rayleigh scatters, and thus, their reflectivity factors should be frequency independent. Furemore, only the cases where there was no precipitation below were selected.**

3   Why did you use the gamma size distribution model (Page 8) rather than directly using PIP observed size distributions expressed in snowflake size bins?

**REPLY. Thanks for the suggestion. We have added Figure 13 (a) that shows PSD for the snowfall case of 12 February 2014 and (b) PSD for snowfall case of 15/16 February 2014, to better clarify the variable shape of experimental PSDs derived from PIP measurements. In Figure 13 (a)-(b) red points are representative of the normalized PSD measured by PIP, dashed black line represents the normalized**

**estimated Gamma-PSD in Equation 4 and green line is the last one truncated at the maximum value of 2.5 multiplied by the median diameter D_{0}. Then we have used the Gamma-PSD instead of PIP measured one because the modelled distribution seems more regular in terms of general trend (as is visible in Figure 13). Only to clarify we show three scatter-plot in Figure B, one for each band, exploiting the comparison between the Gamma-PSD and the measured one. We can conclude looking at Figure B that the mean errors are decidedly small.**
**In the revised paper we have clarified this point on page 11 line 13-21.**

[Figure]

Figure B

4   It would be helpful if, for each frequency, the authors provide figures showing your best Ze-S relations (given in bold font in Table 3 for individual snowfall events) and some previous relations from literature. You cite a number of such relations for W and Ka-bands. For X-band also there have been a fair amount of previous studies (for example, Boucher and Wieler Journal of Climate and Applied Meteorology 1985, p.68; Fujiyoshi et al. JAM 1990, p. 147; Matrosov et al. JTECH 2009, p.2324; Huang et al., C-band, JTECH 2010, p. 637).

**REPLY. Thanks for the useful suggestion.**
**In the revised paper we have inserted a "new" Table 4 with the literature based Ze-S relations for X-, Ka- and W-band. We have also clarified the relation between our Ze-S relations and the relations from literature by adding an extended comment at Table 4 on page 12 line 26-33.**

5   It would be interesting to know if Ze-S relations derived for the IKA C-band frequency would be much different from those at X-band?

**REPLY. We have omitted to use the IKA C-band frequency both because is not collocated in the same BAECC field station but at 64 km west from Hyytiälä in Ikaalinen and also because the IKA radar acquired RHI scans. For the Ze-S relations derived for IKA C-band we remind to Table 3 (replaced below this comment) of von Lerber et al., 2017.**

TABLE 3. The prefactors and exponents of the $Z_e = a_{zs}S^{b_{zs}}$ relation during BAECC SNEX 2014, with $Z_e$ in millimeters to the sixth power per meter cubed and $S$ in millimeters per hour.

| Date (UTC) | $a_{zs}$ | $b_{zs}$ |
| --- | --- | --- |
| 2100 31 Jan–0600 1 Feb | 52.5 | 1.29 |
| 1000–1600 1 Feb | 143.4 | 1.41 |
| 1600–1900 2 Feb | 102.3 | 1.19 |
| 0400–0900 12 Feb | 160.0 | 1.65 |
| 2100 15 Feb–0200 16 Feb | 114.3 | 1.32 |
| 1600 21 Feb–0330 22 Feb | 146.5 | 1.30 |
| 0500–0700 15 Mar | 143.2 | 1.44 |
| 0800–1900 18 Mar | 290.9 | 1.41 |
| 0000–2000 19 Mar | 781.8 | 1.52 |
| 1600–2350 20 Mar | 87.3 | 1.61 |

v Specific comments:

1 Page 5 line 16: It is stated that ARM radar measurements were corrected for attenuation. Is it attenuation due to accumulated snow on the radome or attenuation in falling snow?

**REPLY. We have applied the atmospheric Millimeter-wave Propagation Model (MPM) that predicts attenuation, delay, and noise properties of moist air at frequencies up to 1000 GHz (Liebe, 1985). The attenuation due to the radome or attenuation in falling snow has been avoided by making a sky-noise analysis, as written in Section 2.3 (now, 2.4). When there are sudden jumps in the sky-noise temperature, it means that the increase of the surface temperature may be responsible for the snow melting and then for the radome attenuation. We have avoided all data in which these jumps were present.**

**In the revised paper we have clarified this point in Section 2.4.**

2 It is not clear if in your modeling you assumed the preferential orientation of the particles (Page 8, lines 1-5) or random orientation (Page 9, lines27-31). I do not understand your term "randomly orientated particles at fixed orientation". Please clarify.

**REPLY. The scattering database for rimed snowflakes by Leinonen and Szyrmer (2015) is used in our work. Leinonen and Szyrmer (2015) have achieved preferential alignment of snowflakes as follows:*"To simulate the partial horizontal alignment of snowflakes in the atmosphere, the shortest principal axis of each aggregate is aligned at a normally distributed random angle, with a mean of 0 and a standard deviation of 40"*. Therefore, both soft-spheroid and complex particles are preferentially aligned horizontally. However, their orientation angle distributions and, probably, aspect ratios do not necessary match. It is possible that the soft-spheroid model needed to fit radar observations does not represent exactly geometrical properties of snowflakes. It is also possible that the complex snowflake model is not physically correct. From the radar remote sensing perspective, if both models are consistent with the radar observations then both particle models are correct.**

**In this study we are introducing one of the methods to judge applicability of different scattering models. Of course, the present dataset is limited and more studies in this direction is needed. We have also added more explanation on Section 3.2 and 3.3.**

3  Fig. 8: What coefficients are shown in Fig. 8? Are those corresponding to the dashed black lines in Figs. 3-7? Or something else?

**REPLY. In the revised paper we have changed Figure 8 in Figure 7 having changed also the dataset separation in lightly, moderately rimed and heavily rimed snow. Now Figure 7 shows the coefficients from Table 2 and 3.**

4  Can you provide in Table 2 coefficients corresponding to the dashed black lines in Figs. 3-7?

**REPLY. Thanks for comment to improve clarity of the captions. We changed both Table 2 and Figures 3-7 now corresponds to Figures 4-6. Then now coefficients in Table 2 correspond to the dashed black lines in Figures 4-6.**

5 How did you obtain Dmax from the disk equivalent PIP measurements of Ddeq ?

**REPLY. See reply above to the Minor Comment n. 6 of the Reviewer n. 1.**

6  Fig. 9. What is D0 in this figure? Is it the same as given by eq. (6)?

**REPLY. Thank you to highlight the missed and wrong definition. In eq. (6) we missed to define D0, Veq as the median volume diameter obtained from Dmax, now we have added all the definitions.**

7 Page 4 line 23: mm of water?

**REPLY. Yes, we have indeed missed to add that the weighing precipitation gauge measured mm of water. We have added it in the revised paper.**

8  Radar calibration: As the IKA radar has a vertical resolution of about 1 km at the ARM site (~1 deg @ 64 km) did you averaged vertically ARM radar measurements in vertical to match this resolution?

**REPLY. Yes, to reduce the beam mismatch and to facilitate comparison to the ground-based sensors, all the radar data are averaged to 5-minutes.**
**In the revised paper we have clarified this point on page 7 line 18-21 and in Section 2.**
* * *
–

**RC4:** 'Review AMT-2017-485', Anonymous Referee #4, 31 Jan 2018

*This manuscript presents a very thorough comparison of snowfall measurements conducted at X, Ka and W radar frequency, with the interesting idea of identifying an optimal aspect ratio for each of the frequencies under investigation. The research topic is important, mostly but not only because of the upcoming launch of EarthCare. The manuscript is well written and easy to read. I therefore recommend publication after a few minor corrections.*

    v  Major comment: My only major comment is about the classification of snow, as either fluffy or rimed. Further details should be given about how this distinction is made, and propose for example some shape descriptors to discriminate the transition. This appears as the only major subjective choice to be motivated. I suggest a piece of literature on the subject: "Solid hydrometeor classification and riming degree estimation from pictures collected with a Multi-Angle Snowflake Camera", by Praz et al, AMT 2017. In this study, the authors presented a classification method that tried to be as much as possible in line with the classification of Magono and Lee (1966).

**REPLY. See reply above to the Comment n. 1 of the Reviewer n. 1.**

**The resolution of PIP instrument is coarse compared to MASC and quantitative classification of riming degree cannot be achieved with the same accuracy than in Praz et al.. In this study the descriptor for riming is performed based on radiometer measured liquid water path (LWP).**

    v  Minor comment:

        1   Page 4: Could you add a sentence summarizing the possible limitations/error sources of PIP? (i.e. beef up the final sentence about the wind)

        **REPLY. The revised text states the sizing error because of the blurring in line 6 (Page 4) and the minimum threshold for velocity in line 10 (Page 4) and size resolution in line 1 (Page 4). The limitations of observing the particle from a single projection is stated also in Section 2.1. To avoid repetition between the manuscripts we have published concerning the measurements during BAECC campaign we have stated that the more precise description of the uncertainty can be found in Tiira et al. 2016, von Lerber et al. 2017.**

        2   Page4, Line 14: add the percentage of "rejected" particles for this specific campaign, if applicable

        **REPLY. The fit of observed v(D) and retrieved m(D) is used, therefore the amount of rejected particles are not known. There is a threshold of 30 observed particles during the 5-minute period in order to compute the fit. Typically during the 5-minute period 10^3 particles are observed. The error analysis of the m(D) is discussed more in detail in von Lerber et al. 2017. We have added two sentences on page 4 line 16-18.**

        3 Page 5, Line 24: add an error measure (standard deviation) of such intercomparison

        **REPLY. Standard deviation for the 15 February 2014 at 17:13 UTC where**

**calibration is performed within in the most stable height interval between 4 and 6 km is: 1.04 dB (C Band), 1.14 dBZ (X Band), 1.28 dB (Ka Band), 1.28 dB (W Band).**

4 Page 7, line 5: could you elaborate also in term of sampling volume sizes, other than time?

**REPLY.  The radar sampling volumes are not exactly the same, but similar. Since we are not performing any direct comparison of radar observations, the exact matching of radar volumes is not necessary.  The main discrepancy is in sampling volumes between radar and PIP observations. Then PIP sampling volume depends on particle size and fall velocity. For the 5-min observation time it is about 1 m^3 for a snowflake with falle velocity of 1 m/s. It is still much smaller than the radar volume, but this is the best we can do.**

5 Page 8, Line 15: as a curiosity, did you perform any evaluation about the goodness of fit?

**REPLY. RMSE and NRMSE can be considered a measure of the goodness of fit, but more details will be inserted in a proceeding paper at ERAD 2018 mostly concerning the fit evaluation.**

6 **Typos** Page 2, l.35: typo "from from"

**REPLY. Done.**
* * *
_

**RC5:** 'Comments on "Snowfall retrieval at X, Ka and W band: consistency of backscattering and microphysical properties using BAECC ground-based measurements"', Anonymous Referee #5, 07 Feb 2018

*In this manuscript, the authors use collocated measurements of triple frequency vertically pointing radar measurements of snowfall with surface PIP PSD measurements. Using these collocations, they evaluate T-matrix method (TMM) simulations of snowfall for different snowfall types (fluffy, rimed) to determine the parameterizations that lead to the closest matches to measured reflectivities at different wavelengths. There are few studies available that directly compare the differences in reflectivity-snowfall relationships at three frequencies, and fewer still that do so with measured data. This paper could be a valuable contribution towards the effort to find simple calculations for the complex relationships between snowflake PSDs and reflectivity, but the result are based on an ambiguous definition of aspect ratio that appears both subjective to the radar and objective as a particle property (explained in the comments). With some clarification on the language, I would support the publication of this article.*

    v  Major comments:

        1  I'm having trouble understanding how I'm supposed to view a particle's aspect ratio (rs ). On one hand, rs appears to be a real, measurable property of a particle. It is defined by a major and minor axis (page 7, line 24), and different aspect ratios refer to different specified particle geometries (page 7 lines 25 and 28; page 12 line 6). Throughout the paper, however, rs is also defined and used as a variable tuning parameter that can change for a given PSD depending on the radar frequency (page 12 lines 8-10). If rs signifies a particle shape, than the assumption of that shape shouldn't be able to change depending on the radar being used to observe it. If rs is intended as a tuning parameter, the language in the paper should be clear prevent any interpretations that the rs recommended could represent physical particle properties.

          **REPLY. The aspect ratio is the parameter of the soft-spheroid particle model. It may or may not coincide with the measurable particle property. We have modified the text to make this point clearer.**

        2  In Section 2.3, the authors claim "The cross-calibration method is based on the assumption that in the low reflectivity region at the cloud top the small crystals basically scatter in the Rayleigh regime (Hogan et al., 2000). In these regions, therefore, the measured radar reflectivity values from by all millimeter wave radars should match". These values may not match if there is substantial liquid water present, and liquid water is common in snowing clouds (Wang et.al 2014). Liquid water attenuation is very difficult to predict at different frequencies for supercooled liquid water droplets (Kneifel et. al 2014), and liquid water is also very hard to measure, so this attenuation may not be possible to fully address. But it should be discussed and, if possible, estimated.

          ·    Kneifel, S., Redl, S., Orlandi, E., Löhnert, U., Cadeddu, M. P., Turner, D. D., & Chen, M. T. (2014). Absorption properties of supercooled liquid water between 31 and 225 GHz: Evaluation of absorption models using ground-based

observations. Journal of Applied Meteorology and Climatology, 53(4), 1028–1045. http://doi.org/10.1175/JAMC-D-13- 0214.1

· Wang, Y., Liu, G., Seo, E. K., & Fu, Y. (2013). Liquid water in snowing clouds: Implications for satellite remote sensing of snowfall. Atmospheric Research, 131, 60–72. http://doi.org/10.1016/j.atmosres.2012.06.008

**REPLY. For cross calibration only nonprecipitating clouds with no or little supercooled liquid water were used. Since we have used data from the lowest usable range gates, the expected liquid water atte uation should be less than 1 dB. That is why not liquid water or snow attenuation correction is applied.**

v Specific comments:

1 Numerous spelling and grammar errors throughout. Suggest closer proofreading before final submission.

**REPLY. Thanks for the suggestions, we have heavily revised the manuscript by introducing some modifications highlighted in blue text within the revised text. We hope that the overall text revision is helpfully for a clearer understanding of the content.**

2 Is it necessary to include the information on the Pluvio gauge in 2.1? I don't see the data used in any of the figures.

**REPLY. Pluvio gauge has been used to check the snow rate from PIP and to estimate the LWE. We have added it in the revised paper a sentence on page 4 line 23-25.**

**SC1:** 'conclusions need scrupulous revision', Mario Montopoli, 08 Feb 2018

Major comment: The paper and measurements presented are interesting but there is one important conclusion that is not adequately supported by quantitative analysis. In particular, when discussing figure 13, the authors conclude: "... i) looking at the product between the PIP-derived PSD and the radar cross section σ, we note that the TMM-based product is higher than the DDA one for small ice particles and is lower for the larger particles. The latter consideration leads to the conclusion that the soft-spheroid approximation may work rather well for computing radar reflectivity since the errors for larger particles are compensated by those for smaller particles." This could not be true in general. It depends by the extreme of integration in terms of particle's diameter of the quantity σ*N(D) shown in figure 13 on the right side axis. If you integrate between 0 and 2.5 mm you will probably have a sort of compensation effect. This not likely happens if you consider larger integration limits. Unfortunately, the Authors do not report a figure where they show a statistic of N(D) measured from PIP to have an idea of typical show particle's range diameters for the considered case studies. They should add N(D) figure.

**REPLY. Thank you for the consideration. We have forced the conclusion because our final assessment is valid but only related to our dataset. Thank you also to highlight the need to add a figure on N(D). We have integrated the TMM between 0 and 2.5D0 mm and this was not justified indeed.In the revised paper we have added Figure 13 in which we show the difference between the measured N(D), the estimated Gamma N(D) and the estimated Gamma N(D) truncated at 2.5 multiplied for D0. From Figure 13, respectively for (a) 12 February and (b) 15/16 February, it is noted that there is an under-estimation of the PSD for higher diameter.**

Minor comments:

1  why in figure 13, bottom panel Deq extends up to 14 mm whereas in the other panel it is up to 6 mm?

    **Figure 13 (now 12) shows horizontally-polarized cross-section modelled by TMM and DDA but the diameter disk-equivalent used to set up the numerical simulations is from PIP data, then the maximum value of 2.5*DDeq depends from the observed particles. Now we have changed the maximum value at DDeq=6 mm to fixed the scale limit.**

2  why DDA simulations starts from Deq =0.4 mm whereas TMM starts from approximatively 0.05 mm? Differences of σ*N(D) in that range of diameter can play a role.

    **REPLY. As explained in section 3.3 the DDA cross sections are computed by averaging particle properties within each bin of the PIP measured PSD and plotted at the bin center. Minimum bin center was 0.375 mm which is representative of particles with sizes ranging from 0.250 to 0.5 in size. However, thanks to the reviewer suggestion, we have extended the plot of DDA scattering cross section down to particles of 0.125 mm in size as they are plotted for the TMM quantities for an easier comparison.**

[revised manuscript text omitted]